# Observation of interface piezoelectricity in superconducting devices on silicon

Haoxin Zhou [1,2,3], Eric Li[1,5], Kadircan Godeneli [1,2], Zi-Huai Zhang [1,2,3], Shahin Jahanbani[2,3], Kangdi Yu [1,2], Mutasem Odeh[1,2], Shaul Aloni[4], Sinéad Griffin [2,4] & Alp Sipahigil [1,2,3] ✉

The development of superconducting quantum processors relies on understanding and mitigating decoherence in superconducting qubits. Piezoelectric coupling contributes to decoherence by mediating energy exchange between microwave photons and acoustic phonons. Although bulk centrosymmetric materials like silicon and sapphire are non-piezoelectric and commonly used as qubit substrates, the lack of centrosymmetry at interfaces may induce piezoelectric losses. This effect was predicted decades ago but never experimentally observed in superconducting devices. Here, we report interface piezoelectricity at aluminum-silicon junctions and demonstrate it as a significant loss channel in superconducting devices. Using aluminum interdigital transducers on silicon, we observe piezoelectric transduction from room to millikelvin temperatures, with an effective electromechanical coupling factor $K^2 \approx (3 \pm 0.4) \times 10^{-5}\%$, comparable to weakly piezoelectric substrates. Modeling shows this mechanism limits qubit quality factors to $Q \sim 10^4 - 10^8$, depending on surface participation and mode matching. These findings reveal interface piezoelectricity as a major dissipation channel and highlight the need for heterostructure and phononic engineering in next-generation superconducting qubits.

The decoherence of superconducting qubits is attributed primarily to material imperfections[1]. Among these imperfections, decoherence induced by two-level systems—material imperfections with discrete energy levels—poses the most significant limitation for state-of-the-art superconducting qubits[2,3]. When a superconducting qubit couples to two-level systems, the latter act as electromechanical transducers, causing energy exchange between the microwave photons and acoustic phonons[4,5]. However, acoustic phonon radiation is not solely caused by two-level systems. Superconducting qubits fabricated on piezoelectric substrates or coupled to piezoelectric transducers can also dissipate through electromechanical transduction and phonon radiation[6]. This loss channel is generally considered irrelevant because

superconducting quantum circuits are often fabricated on centrosymmetric substrate materials, such as silicon and sapphire, where piezoelectricity is nominally absent in the bulk. However, the defects and interfaces of these materials often result in deviations from their ideal bulk properties[7,8]. In particular, piezoelectricity can arise at the interfaces even when the bulk materials are not inherently piezoelectric[8]. Electromechanical transduction based on charge-transfer-induced Coulomb forces has been demonstrated by intentionally breaking centrosymmetry, such as by introducing impurities[9–12] or by driving a current[13] in non-piezoelectric materials. However, electromechanical transduction could, in principle, exist without introducing these perturbations and could potentially be

[1]Department of Electrical Engineering and Computer Sciences, University of California, Berkeley, Berkeley, CA, USA. [2]Materials Sciences Division, Lawrence Berkeley National Laboratory, Berkeley, CA, USA. [3]Department of Physics, University of California, Berkeley, Berkeley, CA, USA. [4]Molecular Foundry, Lawrence Berkeley National Laboratory, Berkeley, CA, USA. [5]Present address: Department of Electrical Engineering and Computer Science, Massachusetts Institute of Technology, Cambridge, MA, USA. ✉e-mail: alp@berkeley.edu

present in high-quality superconductor-substrate interfaces used in superconducting qubits.

Here, we characterize the piezoelectric transduction in a widely used heterostructure for superconducting qubit fabrication: the aluminum-silicon junction. Crystalline silicon can exhibit complex interface behaviors. First, the distinct environments of atoms near the surfaces can induce surface lattice relaxation or reconstruction[14–17] (Fig. 1a). Theory predicts that lattice relaxation can induce surface piezoelectricity on the sapphire (0001) surface[8,18,19], and a similar effect is expected on silicon surfaces. Second, the aluminum film alters the electronic structure of silicon near the interface, generating metal-induced gap states[20–22]. As undoped silicon has a higher work function than aluminum[23,24], free electrons transfer from aluminum to silicon so that the heterostructure reaches electrochemical equilibrium while maintaining the continuity of the vacuum energy level across the junction[25] (Fig. 1b). Similar to lattice relaxation, the charge transfer leads to an electric dipole at the interface, which can contribute to a piezoelectric response. Both mechanisms can induce piezoelectric interface loss in superconducting qubits by resulting in phonon radiation (Fig. 1c).

## RESULTS

### Piezoelectric transduction at aluminum-silicon interfaces

To study the interface piezoelectricity at aluminum-silicon interfaces, we designed and fabricated aluminum interdigital transducers (IDT) with a finger pitch of 1.05 μm on an undoped silicon (100) substrate. The IDTs electromechanically transduce phonons in a surface acoustic wave (SAW) delay-line configuration (Fig. 1d). We use a split-finger design[26] to suppress the internal acoustic reflection of the SAW due to mass loading (Supplementary Fig. 1h). Two mechanisms contribute to the microwave transmission. The first is the capacitive crosstalk between the transmitter and receiver IDTs. The second, which occurs only in the presence of interface piezoelectricity, is the transduction between microwaves and SAWs.

Figure 1e shows the transmission coefficient measured at room temperature on devices with various separation distances $d$ between the IDTs (Sample A, Supplementary Fig. 1a). We observe oscillations of the transmission coefficient on top of the smooth background around the IDT resonance of $f = 4.55$ GHz. The differing background levels of $S_{21}$ among the traces arise from variations in the crosstalk levels between devices. (Supplementary Fig. 2) The oscillation results from the interference between the slow-propagating surface acoustic waves and the capacitive crosstalk[27]. To give a more direct demonstration of the piezoelectric transduction, we convert the data to the time domain by performing a Fourier transform, as shown in Fig. 1f. This gives an approximate impulse response of the device. For all devices, the transmission coefficient peaks at $t = t_c = 2.5$ ns, which is the microwave propagation delay and is insensitive to $d$. Remarkably, a second peak appears after a longer delay time $t_s$. Unlike $t_c$, $t_s$ is proportional to $d$. Fitting the data by the relation $d = v \cdot (t_s - t_c)$ gives $v \approx 5063$ m/s (Inset of Fig. 1f), close to the 4920 m/s—the literature reported speed of surface acoustic waves on silicon (100) surface without electrical or mechanical loads[28].

To quantitatively study the piezoelectric transduction and SAW propagation, we apply a rectangular filter in the time domain around $t_s$ (or "time gating") and transfer it back to the frequency domain. This procedure removes the contribution from capacitive crosstalk and restores the resonance peak in the frequency-domain response[29], as shown in Fig. 1g. The transmission coefficient at resonance ($S_{21,0}$) decreases with $d$, which can be attributed to the propagation loss of SAW. The relation between the $S_{21,0}$ and $d$ can be fit to an exponential function $|S_{21,0}| = Ae^{-d/2l}$ with fitting parameters $A = 1.96 \times 10^{-6}$ and $l = 0.6$ mm. The latter corresponds to the decay length of SAW at room temperature. The origin of the propagation loss can arise from various mechanisms, including phonon scattering (Akhiezer damping[30]), bulk

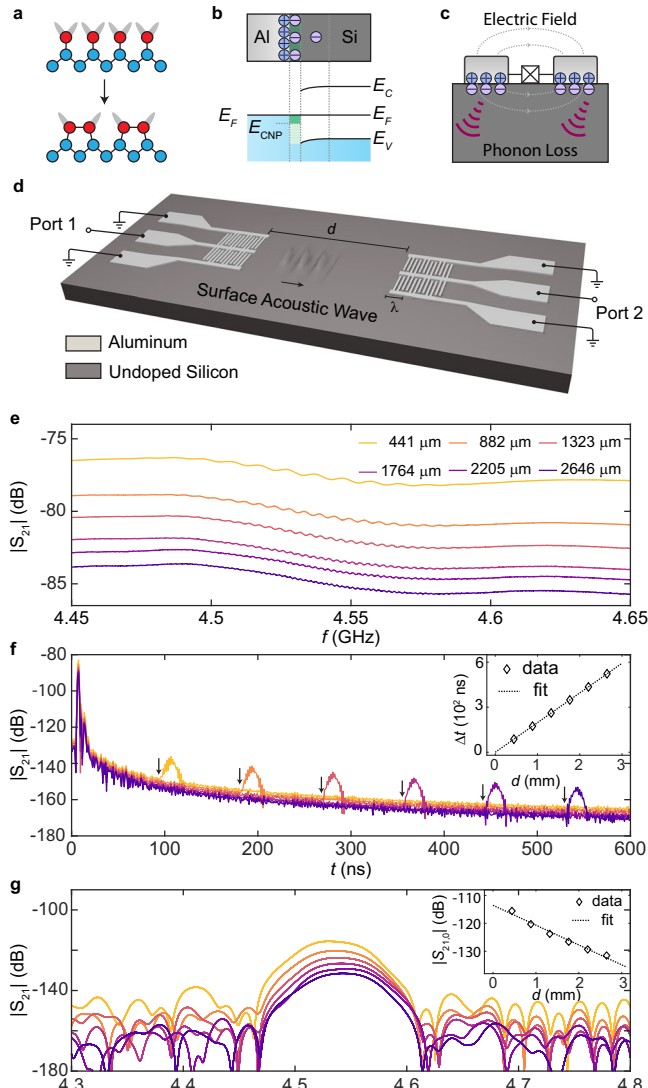

**Fig. 1 | Observation of interface piezoelectricity via surface acoustic waves.** **a** Schematic of the $1 \times 2$ surface reconstruction of silicon (100) surface. Blue spheres: bulk silicon atoms. Red spheres: surface silicon atoms with unpaired bonds (gray). **b** Schematic of the energy band diagram at the aluminum-silicon interface and the formation of interface dipoles. The green region indicates the interface electronic states induced by aluminum. $E_C$ and $E_V$ are the energy of the silicon conduction and valence band edges, $E_F$ is the Fermi energy, and $E_{CNP}$ is the charge neutrality point of the silicon interface states. **c** Interface piezoelectricity induced phonon loss in superconducting qubits. **d** Experiment setup. Aluminum IDTs fabricated on silicon transmit and receive surface acoustic waves. **e** Microwave transmission coefficient $|S_{21}|$ as a function of driving frequency measured for devices with different separation distance $d$ on Sample A (Supplementary Fig. 1a). **f** Time-domain $|S_{21}|$ as a function of delay time $t$. The black arrows indicate the onset time of surface acoustic wave transmission ($t_s$). Inset: $\Delta t = t_s - t_c$ as a function of $d$. Here, $t_s$ ($t_c$) is the onset of transmission mediated by the surface acoustic waves (capacitive crosstalk). Dashed line: Linear fit $d = v \cdot \Delta t$ gives the silicon surface wave velocity $v = 5063$ m/s. **g** Time-gated $|S_{21}|$ as a function frequency for devices with different $d$. Inset: Time-gated $|S_{21}|$ at the electromechanical resonance as a function of $d$. Dashed line: fit. All measurements are conducted at room temperature.

radiation, and diffraction. The observed exponential behavior and the suppression of loss at low temperatures suggest that Akhiezer damping is likely the dominant mechanism, although further experiments are required to reach a definitive conclusion.

We performed the same measurement on devices fabricated with different methods (See Methods section for details) and observed

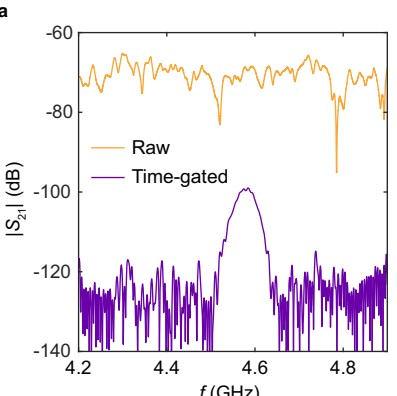

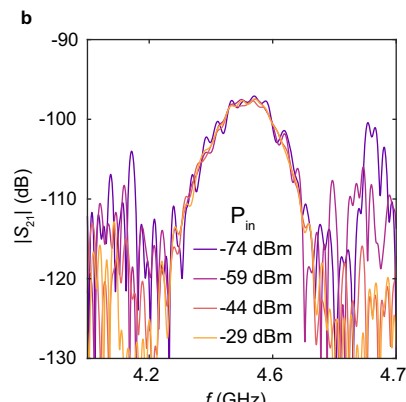

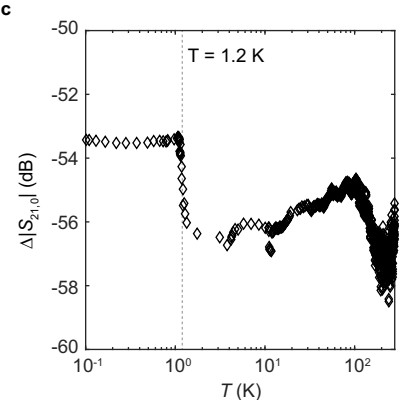

**Fig. 2 | Interface piezoelectric transduction at cryogenic temperatures.**
**a** Orange: Measured transmission coefficient at $T = 30$ mK as a function of frequency on Sample B. Purple: Time-gated transmission coefficient calculated from the orange trace. **b**, The time-gated transmission coefficient as a function of frequency $f$ with different excitation powers on Sample B at $T = 20$ mK. **c**, The time-gated transmission coefficient ratio between Samples B and Sample D (Supplementary Fig. 1d) at the electromechanical resonance as a function of temperature. Gray dashed line indicates the superconducting transition temperature of aluminum.

piezoelectric transduction on all six samples studied (Supplementary Fig. 4). These findings indicate that interface piezoelectricity is robust against fabrication process variations and the presence of native silicon oxide at the aluminum-silicon interface. In the superconducting qubit research community, a common approach to evaluate the quality of superconducting film-substrate heterostructures is by measuring the quality factor of superconducting microwave resonators fabricated on these heterostructures. In our study, we fabricated superconducting resonators using the same heterostructures employed in Samples F and G. (Supplementary Fig. 5a and ref. 31). The superconducting resonators etched from the same aluminum-silicon heterostructure revealed internal quality factors of $Q_i \approx 10^6$ at the single-photon level (Supplementary Fig. 5b, c). These results demonstrate that the heterojunctions used in this study are of comparable quality to those utilized in high-performance superconducting circuits.

## Interface piezoelectricity at millikelvin temperatures

Superconducting qubits operate at cryogenic temperatures, where deviations in electronic and structural properties can influence the piezoelectric response of aluminum-silicon heterostructures. To study the cryogenic response, we fabricated SAW transducers and packaged them for microwave transmission measurements in a dilution refrigerator (Sample B, Supplementary Fig. 1b). Figure 2a shows the microwave transmission coefficient measured at $T = 30$ mK. By applying time-gating to remove the contribution of capacitive crosstalk (which is stronger due to the sample geometry and measurement setup), a clear electromechanical resonance peak is resolved, with the resonance transmission coefficient $|S_{21,0}| \approx -99$ dB. Varying the excitation power by four orders of magnitude does not change $S_{21,0}$ (Fig. 2b). The lack of power dependence distinguishes the observed electromechanical transduction from electrostriction and two-level system response, which are nonlinear. The linear interface piezoelectric surface losses observed here also provide an experimental method to distinguish surface piezoelectric and surface two-level system losses in superconducting devices.

The measured $|S_{21}|_{max}$ value allows us to use an analytical model[32] to calculate the effective electromechanical coupling coefficient $K^2$, the conversion efficiency between electrical and mechanical domains, for the aluminum-silicon IDT (See[27] for more details).

$$K^2 = \frac{1 + (2\pi f_0 C_g Z_0)^2}{2Z_0} \frac{\zeta}{8\gamma C_g f_0 NL} |S_{21}(f_0)|. \qquad (1)$$

Here, $f_0$ is the resonant frequency. $C_g = 318$ fF is the transducer capacitance. $N = 50$ is the number of finger periods. $Z_0 = 50\,\Omega$ is the reference impedance. $\zeta = 1.0836$ and $\gamma = 1.414$ are geometric factors, and $L$ is the propagation loss of SAW. The Akhiezer damping-induced propagation loss is expected to be negligible at millikelvin temperatures due to the lack of thermal phonons. We confirmed this by cryogenic distance-dependent transmission coefficient measurements on Sample C (Supplementary Fig. 1c), as shown in Supplementary Fig. 7. We find that $|S_{21,0}|$ is not sensitive to the separation distance $d$, and therefore $L \approx 1$ at low temperatures. From the results above, we obtain $K^2 \sim 3 \times 10^{-5}\%$ as the effective electromechanical coupling coefficient for aluminum on silicon IDTs. This value is comparable to weakly piezoelectric substrates such as 4H-SiC[29]. The observed electromechanical coupling strength can lead to a significant surface loss channel for superconducting qubits. We analyze the impact on qubits in a later section.

We study the temperature dependence of the transduction efficiency by comparing the microwave transmission coefficient of the aluminum-on-silicon transducers with a control device (Sample D, Supplementary Fig. 1d). This approach allows us to reduce calibration uncertainties due to resistive losses above the superconducting transition of aluminum. The control device has the same geometry except that the electromechanical transduction is mediated by a 200-nm piezoelectric aluminum nitride film (Supplementary Fig. 8b). Since the piezoelectric response of aluminum nitride has a weak temperature dependence[33], the ratio of the microwave transmission coefficients at the electromechanical resonance, $\Delta|S_{21,0}| = |S_{21,0}^{Si}|/|S_{21,0}^{AlN}|$ approximates the temperature dependence of the electromechanical transduction at the aluminum-silicon interface. (See Supplementary Fig. 8 for details of the aluminum nitride film and additional data.) As shown in Fig. 2c, $\Delta|S_{21}|$ exhibits a non-monotonic dependence on temperature. The variation is less than 6 dB within the experimental temperature range and corresponds to a <50% change in the effective piezoelectric coefficient. These measurements demonstrate that interface piezoelectricity also has a weak temperature dependence. An interesting observation is that $\Delta|S_{21}|$ shows a sharp increase as the temperature drops below $T = 1.2$ K, coinciding with the superconducting transition temperature of aluminum (Fig. 2d). This suggests a potential link between interface piezoelectricity and aluminum's superconducting transition, though the mechanism remains unclear.

## Possible microscopic mechanisms of interface piezoelectricity
As previously discussed, both the surface lattice relaxation and work function mismatch-induced interlayer charge transfer can

induce electromechanical transduction. In the latter case, the charge on the silicon side can either exist as thermally activated space charges or be hosted by metal-induced gap states. Depending on the fabrication process, the silicon surface can exhibit different terminations, and a thin layer of silicon oxide may be present on the silicon surface exposed to air. Although this complexity will not affect the symmetry argument (inversion symmetry breaks at material interfaces regardless of the detailed interface configuration), it complicates the microscopic mechanisms of the piezoelectric response.

To gain further insights into the microscopic origins, we study how the transduction depends on silicon surface preparation methods and a DC bias between the aluminum IDT and silicon substrate, as shown in Fig. 3a. We apply a DC bias voltage to an aluminum electrode on the backside of the substrate while keeping the signal and ground electrodes of the IDTs at DC ground. The aluminum back electrode covers the majority of the substrate area and makes ohmic contact to the silicon substrate with a sintering process. (See Methods) We investigate two samples with identical geometry, but one has the native oxide removed by hydrofluoric acid (Sample E, Supplementary Fig. 1e) before the aluminum deposition. Without the bias voltage, the two samples show nearly identical transduction efficiency (Fig. 3b). The presence of the oxide and its thickness may affect the lattice structure near the interface, and therefore, affect the piezoelectric response[8]. By contrast, the built-in electrical potential across the Al/Si junction is primarily set by the work-function mismatch of the two bulk materials in the thin-oxide regime, so the piezoelectric response arising from interface charge transfer is expected to be less sensitive to whether a native oxide is present. In addition, the finite-element analysis discussed later indicates the piezoelectric transduction observed here is significantly stronger than the prediction in ref. 8. The lack of dependence of the transduction efficiency on the presence of native oxide and the larger transduction efficiency suggest that lattice relaxation is probably not the dominant contributor to interface piezoelectricity.

When a bias voltage is applied, Sample A, with the native oxide present, shows enhanced piezoelectric transduction when $V_{Si} > V_{Al}$ and suppressed transduction when $V_{Si} < V_{Al}$ (Fig. 3c, d). This response is consistent with the interface charge transfer picture, as the charge distribution depends on both the work function mismatch and an external electrical potential difference. Furthermore, we used the finite-element method to simulate the charge distribution near an ideal aluminum-silicon junction in response to a bias voltage (Supplementary Fig. 9). The resulting polarity aligns with the experimental results. Interestingly, Sample E, with native oxide removed, does not show a bias-dependence (Fig. 3e, f). A possible explanation is that the electrical potential drop across the accumulation layer near the interface is too low to alter the interface charge distribution when no oxide is present. This observation indicates that the charge that contributes to the piezoelectric response is likely from the surface states of silicon, which are concentrated near the aluminum-silicon interface, instead of thermally activated bulk charges. This is further supported by the temperature dependence of the piezoelectric response. The distribution of the thermally activated space charge is sensitive to temperature[34]. Since we observed a weak temperature dependence (Fig. 2c), the space charge is less likely to be the dominant mechanism. Considering all the observations, the interface charge transfer between aluminum and the metal-induced gap states in silicon is likely the primary contributor to interface piezoelectricity, although further investigation is required for a definitive conclusion.

## Impact on superconducting qubits

Despite the uncertainties in the microscopic mechanism, the experimentally observed interface piezoelectricity will constitute a loss channel for superconducting qubits on silicon.

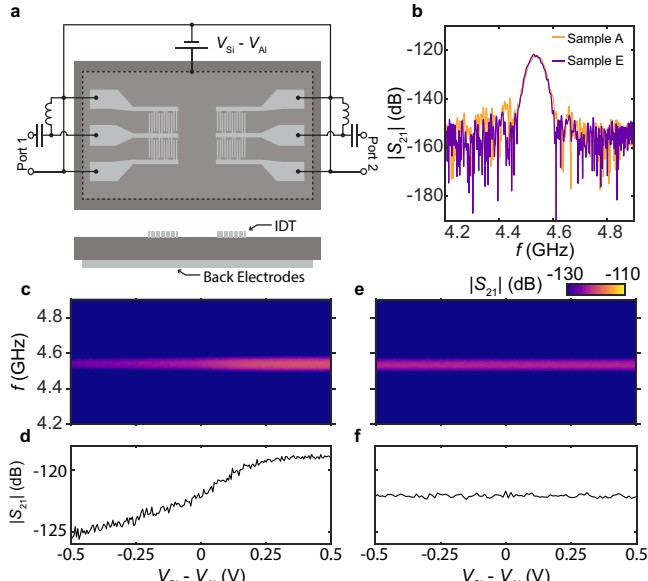

**Fig. 3 | Bias field dependence of the interface piezoelectricity. a** Schematics of the bias-dependent measurement. Top: measurement wiring. Bottom: cross-section of the device. (Also see Supplementary Fig. 12a). **b** Zero-bias time-gated microwave transmission coefficient measured on Samples A and E at room temperature. **c** Time-gated transmission coefficient $|S_{21}|$ as a function of frequency $f$ and bias voltage $V_{Si} - V_{Al}$ measured on Sample A. The separation distance between the IDTs is $d = 1323\,\mu m$. **d** Linecut of the data in panel c at the resonant frequency. **e** Same as **c**, measured on Sample E (Supplementary Fig. 1e), where native oxide is removed. **f** Linecut of the data in panel e at the resonant frequency.

Here, we adopt the circuit model shown in Fig. 4a to estimate the impact of interface piezoelectricity on the lifetime of superconducting qubits[35]. The lossless transmon qubit is characterized by its junction inductance $L_J$ and shunt capacitance $C_g$. Similar to modeling an antenna with an effective radiation impedance, we use the acoustic radiation admittance $Y_a$ to describe the conversion of the qubit electrical energy into acoustic radiation, i.e., the piezoelectric loss. For the IDTs used in the experiment, $Y_a$ can be directly obtained from the effective piezoelectric coupling coefficient $K^{2\,27}$. We use the experimentally observed $Y_a$ in the IDT geometry to calculate the quality factor of a transmon qubit with a shunt capacitor identical to the IDTs used in the experiment (Fig. 4b). The result is shown in Fig. 4c. At the electromechanical resonance, the piezoelectric loss-limited quality factor of the qubit is $Q_{piezo}^{IDT} \approx 7 \times 10^4$, indicating a significant loss channel.

Next, we study the interface piezoelectric loss for more commonly used transmon qubit capacitor geometries shown in Fig. 4d, f. The loss rate induced by interface piezoelectricity in these transmon qubits can differ from the IDT geometry of Fig. 4b due to different surface participation (See Supplementary Fig. 11) and electromechanical mode matching. We use finite-element analysis to estimate interface piezoelectric losses in transmon qubits with 125-fF shunting capacitors with concentric coplanar ("planar", Fig. 4d) and parallel-plate (PPC, Fig. 4f) geometries. The axial symmetry is used to simplify the simulation while keeping the footprint comparable to a typical transmon qubit[36]. The dimensions of the two geometries are shown in Supplementary Fig. 10.

Finite-element analysis requires defining a piezoelectric region within the silicon substrate with a fixed thickness and piezoelectric coupling coefficient. However, our SAW experiment only determines $K^2$. This effective lumped parameter depends on the distributions of electromagnetic and mechanical fields, as well as the properties and thickness of the piezoelectric material. The same value of $K^2$ can arise

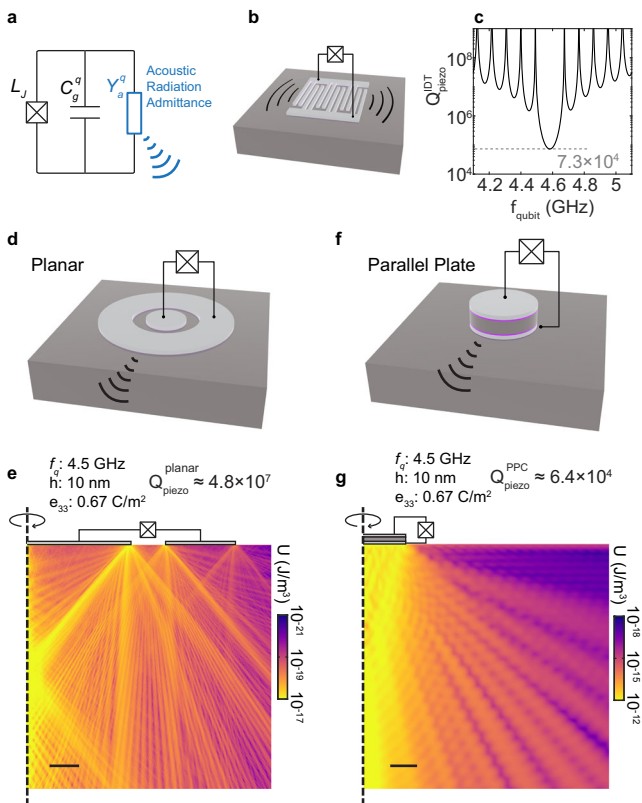

**Fig. 4 | Interface piezoelectric surface loss in superconducting qubits. a** A circuit model of the transmon qubit with interface piezoelectric loss. Electromechanical transduction is modeled by an acoustic radiation admittance $Y_a^q(\omega)$, where $\omega = 2\pi f$ is the angular velocity. **b** Schematic of a transmon with an interdigital capacitor. Phonons are radiated as surface acoustic waves near resonance. **c** The calculated quality factor of a transmon qubit with an interdigital shunt capacitor identical to the IDTs used in the experiment near the electromechanical resonance. **d** A schematic of the transmon qubit with a coplanar shunt capacitor. **e** Simulated spatial distribution of the mechanical energy density within the silicon substrate. The coplanar capacitor is driven at frequency $f = 4.5$ GHz, with the total electrostatic energy equal to that of one microwave photon at $f = 4.5$ GHz. The scale bar represents 50 μm. The positions of the two aluminum electrodes are indicated, though not drawn to scale in the vertical direction. **f** A schematic of the transmon qubit with a parallel-plate shunt capacitor. **g** Same as e, for the parallel-plate capacitor. Scale bar represents 5 μm.

from different combinations of an effective piezoelectric film with thickness $h$ and the piezoelectric coupling coefficient $e_{33}$.

To simulate the qubit loss in different capacitor geometries, we need to know the effective piezoelectric film properties ($h$, $e_{33}$). We use finite-element analysis to find combinations of ($h$, $e_{33}$) that give the experimentally observed $K^2$[27] in an IDT (Supplementary Fig. 6c, d). Next, we use these ($h$, $e_{33}$) combinations to calculate the qubit loss in different capacitor geometries. Figure 4e, g show the radiated mechanical energy density distribution under the qubit capacitors for $h = 10$ nm and $e_{33} = 0.67$ C/m$^2$ in the planar and PPC geometries. The interface piezoelectricity-limited quality factors for the planar and parallel plate capacitor are $Q_{\text{piezo}}^{\text{planar}} \approx 6 \times 10^7$ and $Q_{\text{piezo}}^{\text{PPC}} \approx 7 \times 10^4$, respectively. The large difference between $Q_{\text{piezo}}^{\text{planar}}$ and $Q_{\text{piezo}}^{\text{PPC}}$ arises from their different interface participation ratios. We repeat the same $Q_{\text{piezo}}$ calculations with several different ($h$, $e_{33}$) combinations that match the experimentally observed $K^2$ (Supplementary Fig. 10d). At $f_q = 4.5$ GHz, similar $Q_{\text{piezo}}$ values are found in the range of $Q_{\text{piezo}}^{\text{planar}} \approx 1 - 5 \times 10^7$ and $Q_{\text{piezo}}^{\text{PPC}} \approx 2 - 6 \times 10^4$ for the two geometries. We find a small decrease of $Q_{\text{piezo}}$ for ($h$, $e_{33}$) combinations with large $h$ and smaller $e_{33}$. This observation is consistent with improved piezoelectric

mode matching between the electric and strain fields away from the top surface, which is a strain node. We note that these simulations provide only a rough estimate of the impact of interface piezoelectricity on the performance of superconducting qubits. In addition to the choice of the effective piezoelectric layer thickness, the effective electromechanical coupling coefficient $K^2$ can vary depending on the specific acoustic modes. However, this variation is typically within the same order of magnitude[37] and does not significantly affect the overall order-of-magnitude estimation.

## DISCUSSION

These results establish interface piezoelectricity as a newly identified loss channel for superconducting qubits. State-of-the-art superconducting qubits on silicon have recently achieved quality factors of $Q \approx 7.7 \times 10^6$[38]. Our predicted interface piezoelectric surface loss limit of $Q_{\text{piezo}}^{\text{planar}} \approx 6 \times 10^7$ is already close to the state-of-the-art values. These suggest that interface piezoelectric surface losses could already be significant in state-of-the-art superconducting qubits. In addition, unlike TLS losses, the piezoelectric loss is a linear effect and does not decrease at high power. This may explain the saturated high-power quality factors of resonators[38], including our results shown in Supplementary Fig. 5.

The qubit loss induced by interface piezoelectricity originates from the heterostructure material properties of the qubits rather than material disorders such as defects or amorphous interfaces. As such, even high-quality metal-substrate interfaces will suffer from this loss mechanism. Therefore, delicate engineering at the device level is essential to mitigate this loss mechanism. These might include shielding qubits from the resonant phonon bath using phononic metamaterials[39,40], and interface charge transfer engineering by choosing different qubit heterostructures.

On the positive side, the observed reasonably strong and process-insensitive transduction opens up the possibility of realizing electromechanical quantum transduction with the interface piezoelectricity. Current approaches for electromechanical transduction require either piezoelectric materials[41,42] or static charge induced by an externally applied voltage[43,44], which increases device fabrication and biasing complexity and noise. Utilizing interface piezoelectricity may eliminate both problems, allowing high-performance electromechanical quantum transducers to be fabricated using a relatively simple and industrially compatible process. Finally, metal-oxide-semiconductor structures could also enhance the charge transfer and transduction strength using large biases for classical transducer and filter applications.

## METHODS

### Sample Fabrication

Samples A, B, C, E, F, and G (Supplementary Fig. 1) were fabricated on float-zone intrinsic (100) silicon substrates with room temperature resistivity $> 10^4 \Omega$ cm. The aluminum films, unless noted otherwise, are deposited on the (100) surface of an undoped silicon substrate. Sample D was fabricated on substrates obtained from Kyma Technologies, which have a 200 nm aluminum nitride film on float-zone intrinsic (111) silicon wafers with room temperature resistivity $> 10^4 \Omega$ cm.

Sample A was fabricated by first depositing a 100-nm aluminum film on the backside of the silicon chip using an electron beam evaporator. The chip was subsequently annealed at 420 °C in a forming gas atmosphere to create Ohmic aluminum-silicon contacts. Finally, the IDTs were formed by depositing a 50 nm aluminum film on the front side of the chip with an electron-beam-lithography-defined mask in an electron beam evaporator, followed by a lift-off process. The chip was exposed to an oxygen plasma before the aluminum deposition to clean the aluminum-silicon interface.

Samples B and C were fabricated following a two-step lift-off process. The IDTs were first formed following the same process for

Sample A. The coplanar waveguides were then formed by depositing a 100 nm aluminum film on the chip with a photolithography-defined mask in an electron beam evaporator, followed by a lift-off process. The sample was wire-bonded to a printed circuit board for cryogenic measurements.

Sample D was fabricated following the same process as Sample A on the aluminum nitride substrate.

Sample E was fabricated following a lift-off process similar to that for Sample A. The difference is that the sample was dipped in a 10:1 buffered hydrofluoric acid solution for 16s to remove the surface oxide after the oxygen plasma treatment.

Sample F was fabricated using an etch process. The substrate was first extensively cleaned with piranha solution, diluted hydrochloric acid solution, and 25:1 hydrofluoric acid solution. A 50 nm aluminum film was then deposited using a DC magnetron sputtering system. Finally, the IDTs were formed by electron beam lithography and a plasma etch.

Sample G was fabricated following a two-step etch process. The substrate cleaning and aluminum deposition process was identical to those for Sample F. Afterwards, electron beam lithography and chlorine-based plasma etch (first etch step) were performed to form the IDT fingers and other fine features. Finally, photolithography and a tetramethylammonium hydroxide-based wet etch (second etch step) were performed to form the coarse feature, including the waveguide.

The information of Sample A-G is summarized in Supplementary Table. 1.

The microwave resonator was fabricated with an etch process. After the substrate cleaning and aluminum deposition process identical to those for Sample F, the coplanar waveguide and the microwave resonator were formed by photolithography and a wet etch.

## Measurement

Room temperature measurements were performed using a radio-frequency probe station and a vector network analyzer (VNA). The microwave power delivered to the end of the probes was 15 dBm. The bias-dependence measurements were performed using radio-frequency bias tees. The DC bias voltage was applied using a source meter. The cryogenic measurements were performed in a dilution refrigerator. The power delivered to the wire bonding pads of the devices was −29 dBm unless otherwise indicated. A high-electron-mobility transistor (HEMT) is used to amplify the transmitted signal, while circulators are used to isolate the reflection from the HEMT to the device (See Supplementary Fig. 12b). The power-dependent quality factor of the microwave resonator was obtained by measuring the transmission coefficient of the device in a dilution refrigerator using a VNA. The details are described in ref. 31.

## Data processing

All microwave measurements were performed using continuous-wave excitation. The time-domain responses in Fig. 1f and Supplementary Figs. 4 and 7b were obtained by applying an inverse fast Fourier transform to the complex $S_{21}$ spectra, with a Hamming window applied to suppress spectral leakage.

To extract the time-gated $S_{21}$, the inverse fast Fourier transform was first performed without windowing. The peak corresponding to SAW-mediated transmission was identified, and a temporal bandpass filter was applied to retain only the signal within a window centered on this peak, setting the remainder to zero. A subsequent fast Fourier transform yielded the time-gated $S_{21}$ in the frequency domain. The gating procedure is illustrated schematically in Supplementary Fig. 3a. Although the size of the time window is a free parameter, the results, as shown in Supplementary Fig. 3b, c, are largely insensitive to the exact window width provided that the SAW-mediated peak is fully included and the capacitive crosstalk peak is excluded.

## Data availability

The data generated in this study have been deposited in Zenodo at https://zenodo.org/records/17620739.

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

## Acknowledgements

We thank Rohin Tangirala for experimental assistance, Francois Leonard for fruitful discussions, as well as Robert Schoelkopf and Srujan Meesala for valuable feedback on the manuscript. This work was primarily funded by the U.S. Department of Energy, Office of Science, Office of Basic Energy Sciences, Materials Sciences and Engineering Division under Contract No. DE-AC02-05-CH11231 within the Phonon Control for Next-Generation Superconducting Systems and Sensors program (KCAS23) for device design, modeling, measurements, imaging, and theory. Additional support was provided by the Air Force Office of Scientific Research and the Office of Naval Research under Grant No. FA9550-23-1-0333 (device fabrication and lithography development), by the U.S. Department of Energy, Office of Science, Basic Energy Sciences, Materials Sciences and Engineering Division under Contract No. DE-AC02-05CH11231 within the Quantum Coherent Systems Program KCAS26 (supporting microwave resonator measurements), and an unrestricted gift from Google (transducer characterization). The devices used in this work were fabricated at UC Berkeley's Marvell Nanofabrication Laboratory.

## Author contributions

A.S. conceived and supervised the work. H.Z., K.G., and S.J. fabricated the devices with assistance from S.A.. M.O. Z.Z., H.Z., and K.G. built the measurement instruments. H.Z. and S.J. performed the measurements. H.Z., K.Y., and A.S. analyzed the data. E.L., H.Z., K.Y., and S.G. performed theoretical modeling and simulations. H.Z. and A.S. wrote the manuscript with the input from all authors.

## Competing interests

The authors declare no competing interests.
