## [Transparent Peer Review file · Nature Communications]

Observation of Interface Piezoelectricity in Superconducting Devices on Silicon

Corresponding Author: Professor Alp Sipahigil

Version 1:

Reviewer comments:

Reviewer #1

(Remarks to the Author)

The manuscript presents an in-depth study of piezoelectricity as a loss mechanism in superconducting qubits, focusing on Al on Si devices. A comprehensive set of experiments using interdigital transducers (IDTs) explores several parameters, including input power, separation distance, bias voltage, and fabrication processes. The authors convincingly attribute the observed losses to piezoelectricity and in particular to interface charge transfer between aluminium and metal-induced gap states in the silicon substrate.

The systematic approach to isolating different parameters in the experimental design is commendable and leads to a well-supported conclusion. The finding that interface charge transfer plays a central role in piezoelectric losses is important, particularly given that the quality factors of state-of-the-art transmon qubits are now approaching the same order of magnitude as this study predicts for these intrinsic losses. The implications for materials and device engineering in superconducting quantum circuits are significant and timely.

Although some aspects of the methodology fall slightly outside my direct area of expertise, the experimental techniques and data interpretation appear sound and in line with standard practices in the field. The reporting appears to be sufficiently detailed to allow for reproducibility.

I found two errors that should be addressed prior to publication.

- Fig. 3e – This should be sample E
- Fig S7b – x-axis should be in nm?

In summary, this is a high-quality study with significant implications for the field of superconducting qubits. Subject to correction of the noted errors and minor clarifications, I recommend publication.

Reviewer #2

(Remarks to the Author)

In this manuscript, the authors report the observation of a surface acoustic wave (SAW) at the interface between an aluminum thin film and a silicon bulk substrate and attribute it to the signature of piezoelectricity. This observation is made using microwave transmission measurements on interdigitated transducers (IDTs). The authors then simulate the impact of piezoelectricity on the quality factor of qubit capacitor pads with two different geometries.

While intrinsic piezoelectricity is not expected in centrosymmetric materials like silicon, artificial piezoelectric effects have been reported in oxidized silicon surfaces (e.g., *Phys. Rev. Applied* 14, 034008 (2020)) and porous silicon (e.g., *Appl. Phys. Lett.* 88, 111905 (2006)). Enhanced piezoelectricity has also been observed in PZT/Si structures and attributed to interface disorder. Considering these prior studies, the present work does not, in my view, constitute a significant advance and novelty in the understanding of piezoelectricity at the Si/Al interface.

Furthermore, the suggestion that piezoelectricity can limit qubit coherence is not a novel idea. Prior theoretical and experimental works have examined piezoelectric coupling as a decoherence mechanism in superconducting qubits (e.g., J. M. Martinis, *Quantum Inf. Process.* 8, 81–103 (2009); L. B. Ioffe, *Phys. Rev. Lett.* 93, 057001 (2004)). On the other hand, state-of-the-art fabrication protocols explicitly aim to remove dielectric oxides (and other defects) at the metal-substrate, substrate-air, and metal-air interfaces to mitigate known TLS losses. Thus, even if the reported effect proves valid, it may not be representative of the primary loss mechanisms in carefully engineered, high-coherence qubits. As such, the broader

scientific significance of this study appears limited within the context of the superconducting qubit community.

More importantly, the attribution of the observed SAW to interface piezoelectricity is not sufficiently substantiated. The manuscript lacks in-depth analysis or material characterization to confirm the proposed mechanism. Such evidence is crucial for publication in a high-impact journal.

Below are a few technical comments:

1. The SAW signal in Fig. 1(e) is only evident for devices with narrow electrode spacing. If the SAW is also present in wide-spacing devices, the data should be replotted or processed differently to make the signal visible.
2. Please clarify which trace is the reference for the 75 dB vertical offset. After applying time gating and converting back to the frequency domain (Fig. 1g), all S21 traces show a similar background of ~ -160 dB. This background should correspond to one of the unshifted curves in Fig. 1(e); otherwise, the 75 dB offset needs to be more clearly justified.
3. Can the authors rule out the possibility that the observed oscillations are due to standing waves caused by impedance mismatches in the measurement setup rather than genuine SAWs?
4. The purpose of presenting CPW quality data is unclear. Is the goal to demonstrate film quality, interface cleanliness, fabrication yield, or something else? Please clarify the motivation.
5. Based on Fig. S3, the estimated center trace width of the CPW resonator is approximately 100 μm , which is unusually wide for high-Q resonator measurements. Wider traces are known to be more susceptible to magnetic flux trapping, especially in the presence of residual magnetic fields during cooldown. Please discuss the potential impact of trapped vortices on the internal quality factor, particularly in the single-photon regime. Furthermore, due to the reduced dielectric participation ratio associated with larger trace widths, one would typically expect a higher Q_i . Given the actual trace width of the resonator, the reported quality factor of $1e6$ is not indicative of a high-performance device.
6. Additionally, the frequency shift observed in the S21 response at high power suggests a contribution from kinetic inductance. Could the authors elaborate on whether this effect arises from the Al film properties (e.g., thickness, grain size, disorder), geometric inductance due to the wide trace, or possible fabrication issues? Finally, please clarify how these resonator-related effects—trapped flux, kinetic inductance, and potentially limited film quality—may be related to, or influence, the observation of SAWs. Is it possible that any of these factors contribute to or obscure the SAW signal?
7. The label “original” is ambiguous in Fig. 2(a). If it refers to the raw S21 trace before applying the time-gating filter, it is unclear why the SAW signal is not visible at low temperatures, where it is expected to be more pronounced due to reduced damping. This raises concerns about the reproducibility and robustness of the observation. As a result, I am left wondering how exactly the time-gating filter was implemented. The authors should clearly describe the procedure: What time window was used? How was the gating applied in the frequency domain? Was any temperature dependence considered in the gating parameters? A more detailed explanation is necessary to assess the validity of the filtering method and the reliability of the extracted SAW signal.
8. Based on a DC bias comparison, the enhanced piezoelectric transduction is attributed to the interfacial oxide. This conclusion lacks rigorous support. Discussions such as materials characterization (e.g., spectroscopy, microscopy) to assess oxide composition and thickness, possible chemical doping during BOE/HF cleaning, evidence for interdiffusion between Al and Si, consideration of annealing-induced topographic changes, and evaluation of fabrication consistency across samples should be provided.
9. If the interfacial oxide indeed enhances piezoelectricity, why do the two samples (with and without oxide) show nearly identical microwave responses?
10. The two capacitor types studied (PPC and concentric) are not representative of common superconducting qubit designs, which limits the relevance of the simulations. More relevant geometries would include parallel plate (e.g., IBM, Princeton) or X-mon (e.g., UCSB, Google) designs, which have been associated with higher coherence times. Simulations using these standard geometries would better reflect the potential practical impact of the reported effect.
11. Reference [37] reports a maximum $Q \sim 7.7e6$, not $2e7$ as stated in the manuscript. Moreover, the capacitor design in that reference is a parallel pad geometry. Even assuming the simulated results of $Q \sim 2-6e7$ in this manuscript, piezoelectricity is not currently the dominant loss mechanism in state-of-the-art transmon qubits.
12. The manuscript should be carefully proofread and revised. A few specific issues include:
 - a. Clearly label samples in the DC-bias experiment (e.g., Sample D or E in line 226 and Fig. 3(b) are confused).
 - b. Rename the “microwave/DC measurement” subsection, which currently implies insight into interfacial mechanisms without supporting evidence.
 - c. Each sample should be labeled in Fig. S2; currently, it is difficult to identify them.
 - d. In Fig. S3(b), the frequency range is too wide for high-Q fitting. The resonance fit in panel (c) would benefit from error bars, as low-power curves appear noisy.
 - e. Include a wiring diagram in the Methods section to clarify the microwave measurement setup.

Given the limited significant advances and novelty, insufficient mechanistic evidence, and lack of relevance to mainstream qubit designs, I cannot recommend this manuscript for publication in Nature Communications. After properly addressing the comments listed above, the authors may consider submitting to a more specialized journal where exploratory studies with potential future implications can be appropriately disseminated.

Reviewer #3

(Remarks to the Author)

The authors measure piezoelectric transduction at the interface of aluminum and silicon, which is currently understood to have negligible piezoelectric behavior in bulk. The authors confirm the presence of piezoelectricity by transmitting surface acoustic waves, while ruling out other effects such as electromagnetic coupling, and find that this effect in silicon is comparable in strength to other weakly piezoelectric materials. Not only is this effect demonstrated using common materials deposited in a variety of ways, but this exact material configuration is frequently used in superconducting quantum devices, making this effect very relevant for minimizing decoherence in future devices.

This work confirms the presence of piezoelectricity in an unusual context, so is of interest to the materials science, superconducting quantum device community and greater physics communities as both a phenomenon to explore and a potential tool for future experiments. I find the manuscript compelling, and suitable for publication, provided the following comments are addressed:

1. The main measured and extracted quantities (K^2 , and various Q_{piezo} limits) are presented approximately – it would be beneficial to also provide the uncertainty for these parameters.
2. The authors should clarify if the transmission measurements are performed using a continuous signal (as opposed to a pulsed measurement scheme) and comment on the resolution and accuracy of the time-binned measurement given the experiment bandwidth. For example, is the bandwidth of the feature in Fig. 1g determined by the bin width or the transducer bandwidth?
3. Lines 101-103 claim that exponential attenuation with distance suggests Akheizer damping. Wouldn't any distributed loss mechanism (say, radiation into the bulk) result in exponential attenuation? To demonstrate that the damping is due to the Akheizer effect, one usually has to show temperature or frequency dependence, which the authors do give some evidence of later, but at this point, such a claim seems unsupported.
4. It is unclear what conclusion the authors are drawing in the statement of lines 115-118. From the information presented up to that point, I can conclude that the fabrication method that results in typical resonator Q s also leads to the observation of piezoelectric transduction. However, I think it's premature to claim that the resonator Q s are limited by that effect. Similarly, the sentence on lines 309-313 is a bit vague and confusing. What does it mean that "piezoelectric losses do not saturate"? That they do not decrease at higher powers, as TLSs do due to saturation?
5. Can the authors elaborate on the AIN calibration used for the temperature dependence measurement? Is it performed at each measured temperature point in 2c-d or only at the temperatures shown in Fig. S6? Especially since the aluminum superconducting transition is still visible in the calibrated data, it would be useful to also present the uncalibrated Si and AIN temperature data together as a function of temperature (to match Fig. 2).
6. The authors should clarify the phrase "rapid enhancement near T_c " (on line 183), since transmission shows a mild increase, then drops with respect to increasing temperature. Similarly it would be helpful to label the critical temperature of aluminum in Fig. 2.
7. The authors attribute increased radiation for the parallel plate geometry in part to the larger interface participation ratio. Can the authors elaborate on whether the participation ratio takes into account electrical energy, mechanical energy, or both? In either case, it might be useful for the reader to see the participation ratios.
8. In Fig. S3, an example of a fit used to extract quality factors should be shown to determine whether the phase of the signal was used in fitting, and give the appropriate validity to the quality factor points in (c). Likewise, error bars should be added to (c) to help confirm accuracy of the single-photon measurements of Q_i .
9. The measurements in the text all deal with surface acoustic waves, however the simulations of qubit energy radiation in Fig. 4 appear to be concentrated in the bulk of the material. Can the authors comment on the differences (if any) between surface and bulk transduction specifically for interface-piezoelectric transduction?
10. The authors explore in detail the effects of piezoelectric radiation from large aluminum structures, but this work also poses some very interesting questions about the interface-piezoelectric radiation from a Josephson junction, which can generate very strong electric field gradients in the substrate. Have the authors considered how the radiation loss rates scale for sub-micron structures?

Finally, a few minor notes:

1. The first sentence reads a bit strangely, since "material imperfections" can be interpreted as a source of loss, but "interfaces" is just a place.
2. The sample labels appear to be inconsistent throughout the manuscript, which hinders clarity. In particular sample D is defined as a control AIN device, but is also shown in Figure 3 with native oxide removed (potentially confused with sample E) to measure the effects of the bias field. Similarly, sample B is labelled as oxide-removed in Fig. S2, but this is inconsistent with the descriptions in the Methods section and Table S1.
3. Fig. 1a should either specify what the different colors represent, or use atomic labels. For example are the surface Si atoms hydrogenated or do the gray shapes represent dangling bonds?
4. In Fig. 1d, the surface acoustic wave is very difficult to see in the rendering. Especially if the geometry is not to scale, the relevant parts of the device could be made larger, or the acoustic wave could be artificially colored.

5. Are the curves in Fig. 1e also offset for clarity? The offset (if any) should be specified. To make the vertical axis more useful in both e and f, it would also be useful to clarify which dataset has zero offset and lines up with the vertical axis.
6. In Fig. 3 and relevant text, it should be clarified that the voltage is applied through an ohmic contact. The components or colors in Fig. 3a should also be labelled for clarity.
7. Would Fig. 2c could be made clearer by plotting temperature on a logarithmic scale? This could also allow Fig. 2c-d to be combined.
8. In Fig. S3b, are the plots offset? If so, the offsets should be noted, and the plots should be arranged such that power increases going upwards in the figure. Alternatively for even more clarity, perhaps the vertical axis could be switched to received power.
9. How are the simulated qubit geometry dimensions chosen? The numbers seem very specific.
10. In the methods section (line 378) the "two-step etch" process should be explained.

Reviewer #4

(Remarks to the Author)

Version 2:

Reviewer comments:

Reviewer #1

(Remarks to the Author)

My comments have been addressed. The paper is well written and suitable for a wider audience as targeted by Nature Communications.

Reviewer #2

(Remarks to the Author)

I thank the authors for their detailed response to my previous comments. Below I provide my assessment of the revised manuscript.

R2.2: I agree that the observation of piezoelectricity in an Al/Si bilayer is novel. However, this novelty also determines whether the work is suitable for publication in a journal such as Nature Communications. The authors have performed detailed microwave measurements to support their claims, but all of these observations remain at the macroscopic level. The manuscript lacks in-depth exploration and analysis at the microscopic scale. Even in the section "Possible Microscopic Mechanisms of Interface Piezoelectricity", the discussion and supporting measurements remain macroscopic, without direct microscopic evidence.

R2.3: I find the response to this point unclear. The authors state that "the central focus of our manuscript is not that conventional piezoelectricity limits qubit coherence", yet the manuscript's core message still appears to be that "piezoelectric coupling contributes to decoherence by mediating energy exchange between microwave photons and acoustic phonons", as stated in the abstract. Numerous similar statements and discussions throughout the manuscript continue to link piezoelectricity directly to qubit coherence. I suggest that the manuscript be revised to focus more directly on the origin of piezoelectricity in the Al/Si bilayer, rather than emphasizing its connection to qubit coherence.

R2.4: I do not see experimental evidence supporting the statement: "the interface piezoelectricity we report arises from symmetry breaking at the interface, a structural effect that, in principle, can exist even in ideal, defect-free heterostructures." In practice, a thin oxide layer on the Si surface can form rapidly, and interdiffusion at the Al/Si interface is highly likely. If the authors wish to maintain their claim of interface piezoelectricity arising from ideal symmetry breaking, they must first provide strong evidence that their Al/Si bilayers are nearly ideal and free from such defects.

R2.5: I do not see qualitative improvements in the revised manuscript. As noted in my comments on R2.2 and R2.4, the work still lacks the necessary in-depth analysis or material characterization to confirm the proposed mechanism. Without microscopic evidence it is difficult to distinguish whether the observed effect arises from intrinsic broken symmetry, interfacial disorder, or oxide/diffusion effects. Such evidence is therefore essential for publication in a journal of this level.

R2.10: (1) In the PRX paper cited, dielectric loss was intentionally introduced, and therefore the resonator Q is not representative of intrinsic limits. If the authors claim that their resonators have Q-factors similar to those on doped substrates, then it is important to clarify what other mechanisms could limit their resonator Q. In addition, the APL paper cited analyzes the participation ratio of a qubit in a 3D cavity. Since qubits are subject to additional loss channels beyond those of resonators, a straightforward comparison of quality factors is not appropriate. A discussion of other possible loss mechanisms, such as trapped flux, seems necessary. (2) In Fig. S3, the S21 fits do not accurately capture the dips in the low-power curves (g-k). Since the S21 linewidth directly determines the resonator quality factor, these discrepancies are critical. Consequently, the results presented in Fig. S3(l) should be treated with particular caution and discussed more

carefully.

R2.11: I am not familiar with the nonlinearity induced by TLS as proposed, and I ask the authors to clarify or confirm this hypothesis. It is well known that thin Al films doped with oxygen exhibit kinetic inductance, and a base pressure of 2×10^{-7} Torr is not sufficient to guarantee ultra-high-purity films. Moreover, a high resonator quality factor does not, by itself, confirm film purity, since it reflects only the distribution of dielectric loss contributions. If the authors wish to attribute the observed piezoelectricity to broken symmetry, they must provide direct evidence of film quality and purity.

R2.15: The “parallel-plate” qubits I referred to are those reported in Nat. Commun. 12, 1779 (2021). In the authors’ response, this geometry seems to have been referred to as “planar rectangular.” This geometry represents some of the best reported T1 times in superconducting qubits (300 μ s, and more recently above 1 ms). To my knowledge, the planar circular geometry has not reached such a high T1. It is therefore surprising that in Fig. R1, the participation ratios for planar rectangular and planar circular geometries appear to be the same (though the plot is not very clear to me—please correct me if I am mistaken). I am particularly interested in how the simulations were performed. Since the piezoelectric layer thickness h is only up to 50 nanometers while the metallization features extend over hundreds of microns, the system presents an extreme aspect ratio of order 10^5 . In principle, a full 3D model is necessary to capture the geometry accurately, but such simulations face significant numerical challenges in resolving both scales simultaneously. By contrast, a 2D cross-sectional model may artificially yield identical participation ratios for planar rectangular and planar circular geometries, obscuring possible differences.

For publication in a journal with high impact, the manuscript requires more comprehensive and in-depth analysis to identify and distinguish the microscopic mechanisms behind the observed piezoelectric behavior in Al/Si bilayers. At a minimum, the authors should determine whether the observed effects originate from broken symmetry, interfacial disorder, or film impurities. Without such evidence, I cannot recommend acceptance.

Reviewer #3

(Remarks to the Author)

The responses to my comments in the rebuttal file seem mostly adequate. However, it seems like most of the changes that are mentioned there are not actually implemented in the updated manuscript file we received. For example, response #3.2-3.5, 3.6, 3.9, 3.11, and 3.20. Perhaps there was some mistake and the wrong version of the file was provided? Either way, in some cases I cannot judge if my concerns were completely addressed (for example response #3.20), since the rebuttal refers to additional text that doesn't exist.

Reviewer #4

(Remarks to the Author)

We are pleased to resubmit our manuscript entitled “Observation of Interface Piezoelectricity in Superconducting Devices on Silicon” for consideration in Nature Communications. We sincerely thank the reviewers for their careful reading and insightful feedback, which have greatly improved the quality and clarity of the manuscript.

In response, we have made substantial revisions to the manuscript and the supplementary materials, and we provide a detailed point-by-point response to all comments.

The major changes include

1. Shortened abstract to align with the house style of Nature Communications.
2. Reformatted Fig. 1 and Fig. 2 following the reviewers’ suggestions.
3. Corrected the calculated electromechanical coupling factor from $2 \times 10^{-5}\%$ to $3 \times 10^{-5}\%$.

The simulation results, which depend on it, have also been updated.

4. Added fitting data and error analysis to Fig. S3.
5. Included additional data to Fig. S6 on the temperature dependence measurement from aluminum-on-silicon and aluminum-on-aluminum nitride samples.
6. Added Fig. S9 on surface participation analysis.
7. Added Fig. S10 showing the experimental wiring diagram.
8. Included a new section in the supplementary materials on error analysis of the effective electromechanical coupling factor.

To assist in tracking the revisions, we highlighted the changes in the revised manuscript.

We believe the revised version addresses all of the reviewers’ comments and demonstrates the novelty and significance of our findings. We believe that the improvements meet the high standards for publication in Nature Communications.

Reviewer #1 :

The manuscript presents an in-depth study of piezoelectricity as a loss mechanism in superconducting qubits, focusing on Al on Si devices. A comprehensive set of experiments using interdigital transducers (IDTs) explores several parameters, including input power, separation distance, bias voltage, and fabrication processes. The authors convincingly attribute the observed losses to piezoelectricity and in particular to interface charge transfer between aluminium and metal-induced gap states in the silicon substrate.

The systematic approach to isolating different parameters in the experimental design is commendable and leads to a well-supported conclusion. The finding that interface charge transfer plays a central role in piezoelectric losses is important, particularly given that the quality factors of state-of-the-art transmon qubits are now approaching the same order of

magnitude as this study predicts for these intrinsic losses. The implications for materials and device engineering in superconducting quantum circuits are significant and timely.

Although some aspects of the methodology fall slightly outside my direct area of expertise, the experimental techniques and data interpretation appear sound and in line with standard practices in the field. The reporting appears to be sufficiently detailed to allow for reproducibility.

Response 1.1: We thank the reviewer for their thoughtful summary of our manuscript and for recognizing its significance.

I found two errors that should be addressed prior to publication.

- Fig. 3e – This should be sample E

Response 1.2: Corrected.

- Fig S7b – x-axis should be in nm?

Response 1.3: We confirm that the correct unit should be μm rather than nm. The reason the charge is distributed over such a thick layer is that we are studying intrinsic silicon, where all carriers arise from thermal activation. Due to the low carrier density, a relatively thick region is required to sustain a deviation from charge neutrality and to compensate for the charge transfer induced by the work-function mismatch.

In summary, this is a high-quality study with significant implications for the field of superconducting qubits. Subject to correction of the noted errors and minor clarifications, I recommend publication.

Response 1.4: We thank the reviewer for recommending our manuscript for publication.

Reviewer #2 (Remarks to the Author):

In this manuscript, the authors report the observation of a surface acoustic wave (SAW) at the interface between an aluminum thin film and a silicon bulk substrate and attribute it to the signature of piezoelectricity. This observation is made using microwave transmission measurements on interdigitated transducers (IDTs). The authors then simulate the impact of piezoelectricity on the quality factor of qubit capacitor pads with two different geometries.

Response 2.1: We thank the reviewer for summarizing our manuscript.

While intrinsic piezoelectricity is not expected in centrosymmetric materials like silicon, artificial piezoelectric effects have been reported in oxidized silicon surfaces (e.g., Phys. Rev. Applied 14, 034008 (2020)) and porous silicon (e.g., Appl. Phys. Lett. 88, 111905 (2006)). Enhanced piezoelectricity has also been observed in PZT/Si structures and attributed to interface disorder. Considering these prior studies, the present work does not, in my view, constitute a significant advance and novelty in the understanding of piezoelectricity at the Si/Al interface.

Response 2.2: We appreciate the reviewer's effort in comparing our results with previous publications. While prior studies have investigated the piezoelectric response of engineered silicon materials and heterostructures at room temperature, to our knowledge, there have been no reports of piezoelectricity arising from intrinsic crystalline silicon across a wide temperature range. Our findings provide new insights into the nature of piezoelectricity at silicon interfaces. More importantly, the superconductor/intrinsic silicon interface, such as the commonly used aluminum–silicon interface, is *exactly the same* structure used in superconducting quantum processors. The observation that these structures exhibit a piezoelectric response down to millikelvin temperatures allowed us to establish a new loss mechanism for superconducting qubits from the interface effect of non-piezoelectric materials. None of the previous studies were able to lead to this conclusion. The significant attention and inquiries we have received from the quantum science community following the release of our preprint indicate the relevance and impact of these findings.

Furthermore, the suggestion that piezoelectricity can limit qubit coherence is not a novel idea. Prior theoretical and experimental works have examined piezoelectric coupling as a decoherence mechanism in superconducting qubits (e.g., J. M. Martinis, Quantum Inf. Process. 8, 81–103 (2009); L. B. Ioffe, Phys. Rev. Lett. 93, 057001 (2004)).

Response 2.3: We would like to clarify that the central focus of our manuscript is not that conventional piezoelectricity limits qubit coherence, as we explicitly state in the first paragraph of the manuscript and in reference to Ref. [6]. Rather, the key experimental discovery we present is that piezoelectricity can emerge at the interfaces of nominally non-piezoelectric materials. Piezoelectric loss has not been a major consideration in superconducting qubits. This is because of the assumption that such loss can be avoided by excluding bulk piezoelectric materials from fabrication. Our findings challenge this assumption by demonstrating that interface-induced piezoelectricity can still occur and contribute to decoherence, and therefore, device-level engineering strategies to mitigate such loss mechanisms.

We address the specific examples provided by the referee:

J. M. Martinis, *Quantum Inf. Process.* 8, 81–103 (2009): refers to conventional piezoelectrics (i.e. Aluminum Nitride) as a loss channel, which is indeed widely known and established.

L. B. Ioffe, *Phys. Rev. Lett.* 93, 057001 (2004): Our experiments are the first direct experimental evidence confirming the theoretical predictions of this foundational paper in non-piezoelectric substrates used in cryogenic superconducting circuits.

To highlight the recognition of our work by the area experts, including L. B. Ioffe (the author of the PRL paper cited above), we refer to a recent manuscript by Ioffe and colleagues (Charpentier et al., arXiv:2507.08953v1) citing the arXiv version of our manuscript:

(From Page 1 of Charpentier et al., arXiv:2507.08953v1) for “A considerable body of work has focused on identifying and quantifying the mechanisms responsible for this residual dissipation across various superconducting materials. These mechanisms include dielectric losses [11], non-equilibrium quasiparticles [12–15], magnetic vortices [16], radiative losses [8–10], and, more recently mechanical vibrations [17] or *substrate piezoelectricity* [18, 19].”

Where Refs [18] is L. B. Ioffe, *Phys. Rev. Lett.* 93, 057001 (2004) and Ref [19] is the arXiv version of this manuscript. We hope that the quick adoption of our manuscript as the experimental reference to piezoelectric losses provides evidence on its recognition by the community.

On the other hand, state-of-the-art fabrication protocols explicitly aim to remove dielectric oxides (and other defects) at the metal-substrate, substrate-air, and metal-air interfaces to mitigate known TLS losses. Thus, even if the reported effect proves valid, it may not be representative of the primary loss mechanisms in carefully engineered, high-coherence qubits. As such, the broader scientific significance of this study appears limited within the context of the superconducting qubit community.

Response 2.4: The reviewer’s comments actually highlight the unique nature of our findings. It has been widely believed that superconducting qubit losses primarily originate from impurities, defects, and other forms of disorder, and that such losses can be mitigated through improved material and heterostructure quality. In contrast, the interface piezoelectricity we report arises from symmetry breaking at the interface, a structural effect that, in principle, can exist even in ideal, defect-free heterostructures.

In our experiments, we carefully considered the potential influence of interfacial silicon oxide and impurities. To address this, we investigated samples fabricated using multiple processes. Notably, for samples E and F, the native oxide was removed prior to aluminum deposition, and fabrication procedures followed best practices commonly used in qubit fabrication. The high internal quality factors of superconducting microwave resonators made from the same stack confirm the quality of these devices. Nevertheless, interface piezoelectricity was still observed. The persistence of this effect, regardless of interfacial details, further supports that it originates from structural symmetry breaking rather than from disorder.

More importantly, the attribution of the observed SAW to interface piezoelectricity is not sufficiently substantiated. The manuscript lacks in-depth analysis or material characterization to confirm the proposed mechanism. Such evidence is crucial for publication in a high-impact journal.

Response 2.5: We respectfully disagree with the reviewer’s opinion of our manuscript. As noted by the other two reviewers, we have conducted a systematic and comprehensive investigation of the phenomenon across multiple samples fabricated using different procedures, and over a broad parameter space, including variations in temperature, distance, and bias. Our results provide clear and consistent evidence of piezoelectric transduction at aluminum–silicon interfaces. While we do not yet have a complete microscopic understanding of the mechanism behind this effect, we have outlined several possible explanations at the end of the manuscript. We believe our findings open an important direction for further research, and we are confident that follow-up studies will help deepen the understanding of this phenomenon. We also note that traditional material characterization techniques, such as piezoforce microscopy, don’t have sufficient sensitivity to characterize the effect observed in our SAW measurements, especially at cryogenic temperatures.

Below are a few technical comments:

1. The SAW signal in Fig. 1(e) is only evident for devices with narrow electrode spacing. If the SAW is also present in wide-spacing devices, the data should be replotted or processed differently to make the signal visible.

Response 2.6: The SAW signal is observed in all devices with varying spacings, as shown in Fig. 1f and Fig. 1g. Figure 1e presents the original scattering parameters in the frequency domain as a supplement to Figs. 1f and 1g. As the spacing increases, the phase

of the SAW at the detector becomes increasingly sensitive to the driving frequency due to the large group delay of the signal. As a result, the interference between the direct crosstalk and the SAW leads to higher-frequency oscillations (“wiggles”) in devices with larger spacing. To make this feature more apparent, we have updated the layout of Fig. 1 to make the signatures more obvious in Fig. 1e.

2. Please clarify which trace is the reference for the 75 dB vertical offset. After applying time gating and converting back to the frequency domain (Fig. 1g), all S21 traces show a similar background of ~ -160 dB. This background should correspond to one of the unshifted curves in Fig. 1(e); otherwise, the 75 dB offset needs to be more clearly justified.

Response 2.7: The bottom trace in Fig. 1f is unshifted and serves as the reference; we have added this clarification to the caption of Fig. 1. No artificial offset was applied to Fig. 1e. The reason the “background” levels of the curves do not overlap is due to variations in crosstalk between the two devices (and the measurement probes) for devices with different spacings. To avoid confusion, we have removed the offset of Fig. 1f. After applying time gating, the crosstalk is effectively removed, and the traces in Fig. 1g exhibit a consistent background limited by the measurement noise floor.

3. Can the authors rule out the possibility that the observed oscillations are due to standing waves caused by impedance mismatches in the measurement setup rather than genuine SAWs?

Response 2.8: The measurement setup is impedance-matched up to the launch pads of the interdigital transducers. If the observed oscillations were caused by standing waves between the vector network analyzer and the launch pads, their frequencies would remain unchanged across devices with different spacings on the chip. However, this is not observed in our experiment. Our measurements in Fig 1 unequivocally shows that the oscillations are due to SAWs: (i) Fig 1e, frequency of oscillations changes with on-chip distance, (ii) Fig 1f, the SAW delay in time domain scales linearly with on-chip IDT distance, (iii) The wave velocity that we extract $v=5063\text{m/s}$ (Inset 1f) agrees with previous reports of SAW wave velocity on silicon.

A detailed analysis of these oscillations is provided in Section 1 of the Supplementary Information. Additionally, the time-domain data, which is basically an inverse Fourier transform of the frequency-domain S21, clearly reveals a delayed peak that cannot be attributed to standing waves.

4. The purpose of presenting CPW quality data is unclear. Is the goal to demonstrate film quality, interface cleanliness, fabrication yield, or something else? Please clarify the motivation.

Response 2.9: The purpose is to demonstrate that the interface cleanliness of the samples we observed interface piezoelectricity. We mentioned this in the manuscript Line #110 – Line # 118. We updated the phrases to make them clearer.

5. Based on Fig. S3, the estimated center trace width of the CPW resonator is approximately 100 μm , which is unusually wide for high-Q resonator measurements. Wider traces are known to be more susceptible to magnetic flux trapping, especially in the presence of residual magnetic fields during cooldown. Please discuss the potential impact of trapped vortices on the internal quality factor, particularly in the single-photon regime. Furthermore, due to the reduced dielectric participation ratio associated with larger trace widths, one would typically expect a higher Q_i . Given the actual trace width of the resonator, the reported quality factor of $1e6$ is not indicative of a high-performance device.

Response 2.10: The center trace width is 50 μm instead of 100 μm . While wider traces can indeed be more susceptible to magnetic flux trapping, we believe this discussion is not central to the focus of the manuscript. In addition, the power dependence we observe in Fig. S3c indicates that resonator losses are dominated by TLS, and not trapped vortices. The resonator measurements are included solely to demonstrate the high quality of the material stacks. Moreover, the interdigital transducer (IDT) finger widths used in this study are on the order of 100–200 nm, where trapped flux effects are negligible. As such, the physics of flux trapping is not relevant to the SAW-related observations reported here. Regarding the reviewer’s concern about the quality of the resonators we studied in the work, the impact of surface participation ratio on microwave resonators has been systematically investigated in previous works, including by our group in Zhang et al, Phys. Rev. X 14, 041022 (2024). Our design is based on the surface participation ratio sweep in Zhang et al., and we observe similar Q-factors. Based on Wang et al., Appl. Phys. Lett. 107, 162601 (2015), the quality factor of our resonators matched the typical good results that can be achieved in academic labs.

6. Additionally, the frequency shift observed in the S21 response at high power suggests a contribution from kinetic inductance. Could the authors elaborate on whether this effect arises from the Al film properties (e.g., thickness, grain size, disorder), geometric inductance due to the wide trace, or possible fabrication issues? Finally, please clarify how these resonator-related effects—trapped flux, kinetic inductance, and potentially limited film quality—may be related to, or influence, the observation of SAWs. Is it possible that any of these factors contribute to or obscure the SAW signal?

Response 2.11: We have addressed the film quality concern above. At high power, the small frequency shift could be contributed to by any nonlinear effect, such as kinetic inductance and the interaction with two-level systems. This is common in CPW microwave resonators, and we did not confirm which contribution(s) dominate. For this sample, the 50 nm thick aluminum was deposited with a sputtering tool dedicated to depositing aluminum and its compound with strict requirements on substrate materials, and a base pressure below 2×10^{-7} Torr, ensuring high purity aluminum films. We found that the experimentally measured resonant frequencies match well with the finite-element simulations, where only geometric inductance was taken into consideration. Therefore, kinetic inductance does not play a significant role in this film. This is confirmed in the future by the fact that we observed interface piezoelectricity from room temperature down to millikelvin temperatures. Above the superconducting transition temperature of aluminum, the trapped flux and kinetic inductance are not relevant. We also note that SAW measurements do not use any resonant electrical structures; they are simple transmission lines terminated with an IDT, where the aforementioned effects on resonators don't play an important role.

7. The label “original” is ambiguous in Fig. 2(a). If it refers to the raw S21 trace before applying the time-gating filter, it is unclear why the SAW signal is not visible at low temperatures, where it is expected to be more pronounced due to reduced damping. This raises concerns about the reproducibility and robustness of the observation. As a result, I am left wondering how exactly the time-gating filter was implemented. The authors should clearly describe the procedure: What time window was used? How was the gating applied in the frequency domain? Was any temperature dependence considered in the gating parameters? A more detailed explanation is necessary to assess the validity of the filtering method and the reliability of the extracted SAW signal.

Response 2.12: “Original” refers to the raw S21 trace before applying time-gating. We updated the caption of Fig. 2 to make this clearer. We applied the same procedure to obtain the time-domain response as well as the time-gated frequency domain response. We added a section in the S.I. to provide detailed information. The temperature dependence measurements in this study were carried out in a dilution refrigerator, while the room-temperature measurements were carried out with an RF probe station. As pointed out in Line #128 – Line #129 of the manuscript, the design of the coplanar waveguide, the presence of a PCB chip holder, the fridge wiring, and the electromagnetic environment make the capacitive crosstalk significantly stronger and therefore observation of the contribution from frequency-domain SAW data is challenging. Nevertheless, the signature of SAW transduction clearly and consistently appears in the time domain and time-gated spectrum. The former is simply a Fourier transform of the RAW data.

We again note that electrical circuits for the SAW measurements didn't involve any high-Q structures like resonators. Reducing electrical losses at cryogenic temperatures only has a weak effect.

8. Based on a DC bias comparison, the enhanced piezoelectric transduction is attributed to the interfacial oxide. This conclusion lacks rigorous support. Discussions such as materials characterization (e.g., spectroscopy, microscopy) to assess oxide composition and thickness, possible chemical doping during BOE/HF cleaning, evidence for interdiffusion between Al and Si, consideration of annealing-induced topographic changes, and evaluation of fabrication consistency across samples should be provided.

Response 2.13: The bias dependence of the transduction and its sensitivity to interface treatment is indeed an intriguing observation. We include these results as they offer valuable hints for exploring the origin of the observed interface piezoelectricity. However, we emphasize that this is not the central discovery of our study, and we do not draw definitive conclusions from these measurements.

During interface cleaning, we strictly adhered to a procedure analogous to that used prior to the thermal oxidation step in metal-oxide-semiconductor (MOS) fabrication, which is designed to rigorously exclude unintended dopants. Importantly, only the backside aluminum-silicon interface underwent thermal annealing to form Ohmic contacts for applying bias voltages. The IDT-silicon interface did not undergo any annealing, and as described in the Methods section, not all samples in this study were annealed. Therefore, thermal effects are not relevant to the phenomena we report here.

We would also like to highlight that our fabrication process aligns with established best practices for preparing interfaces in superconducting qubit fabrication, where cleanliness and dopant control are of critical importance. The key message we wish to convey is that such standard fabrication procedures, commonly used in superconducting qubit processor development, can inherently lead to the formation of interface piezoelectricity. This, in turn, may have implications for qubit performance. A detailed investigation of the atomic-scale interface structure and the precise microscopic mechanism behind the interface piezoelectricity is beyond the scope of the present study.

9. If the interfacial oxide indeed enhances piezoelectricity, why do the two samples (with and without oxide) show nearly identical microwave responses?

Response 2.14: We did not claim that the presence of interfacial oxide enhances piezoelectricity. Our observations indicate that the interfacial oxide influences the bias

response, but we do not assert that it strengthens the piezoelectric effect in the absence of a bias voltage.

10. The two capacitor types studied (PPC and concentric) are not representative of common superconducting qubit designs, which limits the relevance of the simulations. More relevant geometries would include parallel plate (e.g., IBM, Princeton) or X-mon (e.g., UCSB, Google) designs, which have been associated with higher coherence times. Simulations using these standard geometries would better reflect the potential practical impact of the reported effect.

Response 2.15: We employed rotational symmetry in our simulations to reduce computational complexity. These simulations are particularly demanding because they require meshing with spatial resolution significantly finer than the acoustic wavelength across the entire mechanical domain to ensure reliable results.

The loss due to interface piezoelectricity is primarily determined by the energy participation ratio of the piezoelectric layer, rather than the precise device geometry. The concentric structure used in our simulations yields an energy participation ratio comparable to that of typical coplanar (We assume this is what the referee means: IBM and Princeton do not use parallel plate geometries in published work) or X-mon geometries and, consequently, results in a similar loss effect. We have added a discussion of energy participation ratio analysis of the geometries we used in the updated manuscript. In addition, we calculated the electrical energy participation ratio of a transmon qubit with rectangular coplanar geometry with the same shunt capacitance, and found that the energy participation ratio is close to the axially symmetric structure, shown in Fig. R1.

To our knowledge, the parallel-plate geometries mentioned by the reviewer are not commonly used in superconducting qubit design. In contrast, concentric geometries are actively employed in qubit design to mitigate loss from material disorder and surface effects—for example, see Martinis et al., *npj Quantum Information* 8, 26 (2022), and Stehli et al., *Applied Physics Letters* 117, 124005 (2020). In fact, we received positive feedback from the qubit community for the choice of the circular geometry.

Fig. R1, Electric participation analysis of transmon qubits with different geometries. a. Schematic geometry of the shunt capacitor of a rectangular planar transmon qubit. The width and length of each pad are $200\ \mu\text{m}$ and $690\ \mu\text{m}$, respectively. The gap between the two pads is $20\ \mu\text{m}$. These dimensions give a capacitance of $125\ \text{fF}$. b. Electric energy participation as a function of effective piezo thickness. Different symbols represent different capacitor geometries.

11. Reference [37] reports a maximum $Q \sim 7.7\text{e}6$, not $2\text{e}7$ as stated in the manuscript. Moreover, the capacitor design in that reference is a parallel pad geometry. Even assuming the simulated results of $Q \sim 2\text{--}6\text{e}7$ in this manuscript, piezoelectricity is not currently the dominant loss mechanism in state-of-the-art transmon qubits.

Response 2.16: We thank the reviewer for pointing this out. Reference [37] indeed reports a quality factor of 7.7×10^6 , and we have updated the relevant paragraphs in the manuscript accordingly.

We would like to clarify that we do not claim interface piezoelectricity to be the sole or dominant loss mechanism in state-of-the-art superconducting qubits. However, improving qubit quality factors remains essential for realizing practical quantum computing. As materials and device structures continue to improve and disorder-related losses are reduced, non-disorder-based mechanisms, such as interface piezoelectricity, will become increasingly relevant and must eventually be addressed.

12. The manuscript should be carefully proofread and revised. A few specific issues include:

a. Clearly label samples in the DC-bias experiment (e.g., Sample D or E in line 226 and Fig. 3(b) are confused).

Response 2.17: We updated the references to the devices studied to avoid confusion.

b. Rename the “microwave/DC measurement” subsection, which currently implies insight into interfacial mechanisms without supporting evidence.

Response 2.18: We updated the title to “Possible Microscopic Mechanisms of Interface Piezoelectricity”.

c. Each sample should be labeled in Fig. S2; currently, it is difficult to identify them.

Response 2.19: Added.

d. In Fig. S3(b), the frequency range is too wide for high-Q fitting. The resonance fit in panel (c) would benefit from error bars, as low-power curves appear noisy.

Response 2.20: The fitting was performed over a 200 kHz window centered around the resonant frequency. We added the fitted curves in Fig. S3 for each power value, as well as the error bars for the fitted quality factors.

e. Include a wiring diagram in the Methods section to clarify the microwave measurement setup.

Response 2.21: We added the wiring diagrams of the measurements as a supplementary figure.

Given the limited significant advances and novelty, insufficient mechanistic evidence, and lack of relevance to mainstream qubit designs, I cannot recommend this manuscript for publication in Nature Communications. After properly addressing the comments listed above, the authors may consider submitting to a more specialized journal where exploratory studies with potential future implications can be appropriately disseminated.

Response 2.22: As our response above, and the positive feedback from the other two reviewers, we do not believe the reviewer’s assessment of our manuscript is justified.

Reviewer #3 (Remarks to the Author):

The authors measure piezoelectric transduction at the interface of aluminum and silicon, which is currently understood to have negligible piezoelectric behavior in bulk. The authors confirm the presence of piezoelectricity by transmitting surface acoustic waves, while ruling out other effects such as electromagnetic coupling, and find that this effect in silicon is comparable in strength to other weakly piezoelectric materials. Not only is this effect demonstrated using common materials deposited in a variety of ways, but this exact material configuration is frequently used in superconducting quantum devices, making this effect very relevant for minimizing decoherence in future devices.

This work confirms the presence of piezoelectricity in an unusual context, so is of interest to the materials science, superconducting quantum device community and greater physics communities as both a phenomenon to explore and a potential tool for future experiments. I find the manuscript compelling, and suitable for publication, provided the following comments are addressed:

Response 3.0: We thank the reviewer for their thoughtful summary of our study, for articulating the significance of our work, and for their positive assessment regarding its suitability for publication.

1. The main measured and extracted quantities (K^2 , and various Q_{piezo} limits) are presented approximately – it would be beneficial to also provide the uncertainty for these parameters.

Response 3.1: We have added an evaluation of the uncertainties in the manuscript.

2. The authors should clarify if the transmission measurements are performed using a continuous signal (as opposed to a pulsed measurement scheme) and comment on the resolution and accuracy of the time-binned measurement, given the experiment bandwidth. For example, is the bandwidth of the feature in Fig. 1g determined by the bin width or the transducer bandwidth?

Response 3.2: The transmission measurements were performed using a continuous-wave signal. The experimental bandwidth is 1 GHz, corresponding to a time-domain resolution of approximately 1 ns. When applying time-gating, we used a 100 ns time window, which corresponds to a frequency resolution of about 10 MHz. The feature shown in Fig. 1g spans roughly a 100 MHz range and is therefore primarily determined by the transducer

bandwidth, although the finite bin width is not entirely negligible. We have confirmed that the peak value of S_{12} is not sensitive to the choice of the time window. Recognizing the importance of these experimental details, we have added the relevant discussion to the Supplementary Information.

3. Lines 101-103 claim that exponential attenuation with distance suggests Akheizer damping. Wouldn't any distributed loss mechanism (say, radiation into the bulk) result in exponential attenuation? To demonstrate that the damping is due to the Akheizer effect, one usually has to show temperature or frequency dependence, which the authors do give some evidence of later, but at this point, such a claim seems unsupported.

Response 3.3: We agree with the reviewer and reached this conclusion via temperature-dependent measurements. We attribute the observed loss primarily to Akhiezer damping because the propagation loss becomes negligible at low temperatures, as shown in Fig. S5b and S5d. Other potential loss mechanisms, such as bulk radiation and diffraction, would not be expected to diminish significantly with temperature, and therefore should remain if they were dominant. Nevertheless, we agree with the reviewer that a more systematic investigation is necessary to draw a definitive conclusion. We have revised the corresponding statement in the manuscript to reflect this.

4. It is unclear what conclusion the authors are drawing in the statement of lines 115-118. From the information presented up to that point, I can conclude that the fabrication method that results in typical resonator Qs also leads to the observation of piezoelectric transduction. However, I think it's premature to claim that the resonator Qs are limited by that effect. Similarly, the sentence on lines 309-313 is a bit vague and confusing. What does it mean that "piezoelectric losses do not saturate"? That they do not decrease at higher powers, as TLSs do due to saturation?

Response 3.4: We apologize for the convoluted presentation of the microwave resonator data. Our intention was not to suggest that the quality factors of the resonators are limited by interface piezoelectricity. Rather, we aimed to demonstrate that the interfaces studied in our work are representative of those commonly used in superconducting quantum circuits, and do not exhibit excessive disorder or degradation that would render them atypical. We revised the relevant paragraphs of the main text of the manuscript to clarify this point.

By "piezoelectric losses do not saturate", we indeed mean that it is a linear effect and does not decrease at higher power. We updated the relevant sentence in the manuscript to avoid confusion.

5. Can the authors elaborate on the AlN calibration used for the temperature dependence measurement? Is it performed at each measured temperature point in 2c-d or only at the temperatures shown in Fig. S6? Especially since the aluminum superconducting transition is still visible in the calibrated data, it would be useful to also present the uncalibrated Si and AlN temperature data together as a function of temperature (to match Fig. 2).

Response 3.5: We performed measurements over a densely sampled temperature range, as shown in Fig. 2d, for both the AlN and Si samples. Due to limitations in temperature control flexibility and precision, the two datasets were not acquired at identical temperature points. To compare them, we applied linear interpolation to the AlN data to match the temperature sampling points of the Si data, and then computed the difference to generate the curve shown in Fig. 2d. We have added the uncalibrated temperature data for both Si and AlN samples, shown in the revised Fig. S6 e - f. A detailed description of the data processing procedure was also added to the caption of the revised Fig. S6.

6. The authors should clarify the phrase “rapid enhancement near T_c ” (on line 183), since transmission shows a mild increase, then drops with respect to increasing temperature. Similarly it would be helpful to label the critical temperature of aluminum in Fig. 2.

Response 3.6: By “rapid enhancement,” we refer to a large magnitude of the temperature derivative $|dS_{21}/dT|$. To avoid confusion, we have revised the wording in the manuscript accordingly. Additionally, we have added a label indicating the critical temperature of aluminum in Fig. 2.

7. The authors attribute increased radiation for the parallel plate geometry in part to the larger interface participation ratio. Can the authors elaborate on whether the participation ratio takes into account electrical energy, mechanical energy, or both? In either case, it might be useful for the reader to see the participation ratios.

Response 3.7: By interface participation ratio, we primarily refer to the electrical energy participation. A higher electrical energy participation indicates a stronger electric field concentrated near the interface, which leads to greater conversion of electrical to mechanical energy, i.e., increased mechanical loss. We have calculated the energy participation ratio for the simulated device geometry and included the results in the revised manuscript as well as Fig. R1 of this document.

8. In Fig. S3, an example of a fit used to extract quality factors should be shown to determine whether the phase of the signal was used in fitting, and give the appropriate validity to the quality factor points in (c). Likewise, error bars should be added to (c) to help confirm accuracy of the single-photon measurements of Q_i .

Response 3.8: We added an example fitting as well as the error bars.

9. The measurements in the text all deal with surface acoustic waves, however the simulations of qubit energy radiation in Fig. 4 appear to be concentrated in the bulk of the material. Can the authors comment on the differences (if any) between surface and bulk transduction specifically for interface-piezoelectric transduction?

Response 3.9: This is a good point. Accurately characterizing the transduction efficiency for different mechanical modes requires detailed knowledge of the electromechanical coupling tensor elements, which unfortunately cannot be fully extracted from surface acoustic wave measurements alone. To address this challenge, we performed simulations where the underlying assumptions about the piezoelectric tensor were varied in Fig. S8. As a first step, we generated sets of effective piezoelectric film thickness and coefficients that matched the experimentally observed K2 measurements. We later used these effective material coefficients to calculate bulk transduction efficiency. Our results are shown in Fig S8, and indicate that bulk radiation losses inferred depend weakly on the assumptions. However, we note that piezoelectric transduction, unlike dielectric loss, is a complicated process that involves mode matching between electrical and acoustic domains, and depends on details such as metal film thickness. A detailed study of how bulk piezoelectric radiation depends on geometry, metal film thickness, etc., is beyond the scope of this study and is something we plan to address in future work.

However, studies of multimode bulk acoustic wave transducers show that while coupling strength can vary between modes, it typically remains within the same order of magnitude, especially for the primary modes of interest. (e.g. Han et al., PRL 117, 123603 (2016)), Therefore, even if there are mode-specific variations, they are unlikely to qualitatively affect our conclusions regarding transduction efficiency. We added a paragraph in the manuscript

10. The authors explore in detail the effects of piezoelectric radiation from large aluminum structures, but this work also poses some very interesting questions about the interface-piezoelectric radiation from a Josephson junction, which can generate very strong electric field gradients in the substrate. Have the authors considered how the radiation loss rates scale for sub-micron structures?

Response 3.10: Thank you for the insightful comments. As noted in the earlier response, piezoelectric transduction depends on details of the structures at the relevant phonon wavelengths of $\sim 500\text{nm}$, and involves wave interference in multilayer structures. Although our present work examines only the aluminum–silicon interface, symmetry breaking at the

aluminum–aluminum-oxide boundary could likewise induce a piezoelectric response. Because the electric field is strongly concentrated across the native oxide, any such effect might contribute appreciably to qubit loss. However, we chose not to speculate in the manuscript and consider this a promising direction for future investigation.

Finally, a few minor notes:

1. The first sentence reads a bit strangely, since “material imperfections” can be interpreted as a source of loss, but “interfaces” is just a place.

Response 3.11: We updated the phrase.

2. The sample labels appear to be inconsistent throughout the manuscript, which hinders clarity. In particular sample D is defined as a control AlN device, but is also shown in Figure 3 with native oxide removed (potentially confused with sample E) to measure the effects of the bias field. Similarly, sample B is labelled as oxide-removed in Fig. S2, but this is inconsistent with the descriptions in the Methods section and Table S1.

Response 3.12: Thank you for pointing out the typos and the unclear description of the samples. We have updated the relevant text and figures to make sure each sample is clearly referenced.

3. Fig. 1a should either specify what the different colors represent, or use atomic labels. For example are the surface Si atoms hydrogenated or do the gray shapes represent dangling bonds?

Response 3.13: Thank you for pointing this out. We added the relevant information to the caption of Fig. 1a.

4. In Fig. 1d, the surface acoustic wave is very difficult to see in the rendering. Especially if the geometry is not to scale, the relevant parts of the device could be made larger, or the acoustic wave could be artificially colored.

Response 3.14: We revised Fig. 1d with the suggestions implemented.

5. Are the curves in Fig. 1e also offset for clarity? The offset (if any) should be specified. To make the vertical axis more useful in both e and f, it would also be useful to clarify which dataset has zero offset and lines up with the vertical axis.

Response 3.15: The offset is present in the original data and arises from variations in crosstalk levels when measuring devices with different separation distances. We have added a supplementary figure and accompanying discussion to illustrate and explain this effect. In addition, we have revised the figure caption to make this explicit. We realized the artificial offset of Fig. 1f is confusing, and therefore removed the offset in the revised manuscript.

6. In Fig. 3 and relevant text, it should be clarified that the voltage is applied through an ohmic contact. The components or colors in Fig. 3a should also be labelled for clarity.

Response 3.16: We updated Fig. 3a and the main text to implement these suggestions.

7. Would Fig. 2c could be made clearer by plotting temperature on a logarithmic scale? This could also allow Fig. 2c-d to be combined.

Response 3.17: We updated Fig. 2c axis to a log scale.

8. In Fig. S3b, are the plots offset? If so, the offsets should be noted, and the plots should be arranged such that power increases going upwards in the figure. Alternatively for even more clarity, perhaps the vertical axis could be switched to received power.

Response 3.18: The plots are offset. We added this information to the caption of Fig. S3 and changed the order of the curves. We believe this eliminated the confusion. We prefer not to plot received power since scattering parameters are more commonly used for this type of plot.

9. How are the simulated qubit geometry dimensions chosen? The numbers seem very specific.

Response 3.19: The numbers are chosen to allow the capacitance of the structures to be close to identical and fall in the regime for a typical transmon qubit. No special consideration was taken in choosing the specific number combinations. We added the description of this procedure in the supplementary information.

10. In the methods section (line 378) the “two-step etch” process should be explained.

Response 3.20: We added the description in the method section.

We thank all referees for their careful review of the manuscript. Below is a point-by-point response to all outstanding questions and comments.

Reviewer #1:

My comments have been addressed. The paper is well written and suitable for a wider audience as targeted by Nature Communications.

Response 1.1: We appreciate the reviewer's positive feedback and recommendation.

Reviewer #2:

I thank the authors for their detailed response to my previous comments. Below I provide my assessment of the revised manuscript.

R2.2: I agree that the observation of piezoelectricity in an Al/Si bilayer is novel. However, this novelty also determines whether the work is suitable for publication in a journal such as Nature Communications. The authors have performed detailed microwave measurements to support their claims, but all of these observations remain at the macroscopic level. The manuscript lacks in-depth exploration and analysis at the microscopic scale. Even in the section “Possible Microscopic Mechanisms of Interface Piezoelectricity”, the discussion and supporting measurements remain macroscopic, without direct microscopic evidence.

Response 2.1: We respectfully disagree with the reviewer’s opinion that novelty in this field requires a fully microscopic account. Our experiments provide clear, reproducible evidence of piezoelectricity at superconducting Al–Si interfaces. While the precise microscopic mechanism remains to be resolved, the relevance to superconducting-qubit performance is unambiguous. This situation is similar to the long-standing case of two-level systems. Despite the absence of agreement on their microscopic identity, phenomenological models have driven effective mitigation strategies and steady progress. Likewise, our work identifies a previously underappreciated loss channel, offers a framework for mitigation, and is attracting research interests on the very microscopic studies the reviewer advocates. As also recognized by the other reviewers, we believe these contributions warrant publication in a high-impact journal such as Nature Communications.

R2.3: I find the response to this point unclear. The authors state that “the central focus of our manuscript is not that conventional piezoelectricity limits qubit coherence”, yet the manuscript’s core message still appears to be that “piezoelectric coupling contributes to

decoherence by mediating energy exchange between microwave photons and acoustic phonons”, as stated in the abstract. Numerous similar statements and discussions throughout the manuscript continue to link piezoelectricity directly to qubit coherence. I suggest that the manuscript be revised to focus more directly on the origin of piezoelectricity in the Al/Si bilayer, rather than emphasizing its connection to qubit coherence.

Response 2.2: The significance of our observations is that the piezoelectric coupling-induced loss is not just limited to bulk piezoelectric materials. People have reported piezoelectricity-induced qubit loss when coupling a superconducting qubit to a piezoelectric transducer, such as those fabricated on piezoelectric aluminum nitride. But this is not a concern since the problem can be simply avoided by not introducing bulk piezoelectric materials. Indeed, most qubits use silicon and sapphire as non-piezoelectric substrates. In our work, we show that even if conventional, and nominally non-piezoelectric materials, such as aluminum and silicon, are used for qubit fabrication, piezoelectric loss can still be present due to our observed interface piezoelectricity. We updated the manuscript to make this point clearer. Investigating the mechanism of piezoelectricity in the Al/Si bilayer is of its own interest, but that is not our central focus for this manuscript.

R2.4: I do not see experimental evidence supporting the statement: “the interface piezoelectricity we report arises from symmetry breaking at the interface, a structural effect that, in principle, can exist even in ideal, defect-free heterostructures.” In practice, a thin oxide layer on the Si surface can form rapidly, and interdiffusion at the Al/Si interface is highly likely. If the authors wish to maintain their claim of interface piezoelectricity arising from ideal symmetry breaking, they must first provide strong evidence that their Al/Si bilayers are nearly ideal and free from such defects.

Response 2.3: We did not intend to suggest that the interface piezoelectricity we observed here arises from symmetry breaking at ideal material junctions. What we would like to point out is the general fact that piezoelectricity does not rely on disorder effects. At interfaces, inversion symmetry is inherently absent, regardless of the presence or absence of oxide, which already allows piezoelectricity from a symmetry perspective. In our experiment, we found the piezoelectric coupling factor is insensitive to the silicon surface preparation method, which is consistent with the statement we made in the introduction.

R2.5: I do not see qualitative improvements in the revised manuscript. As noted in my comments on R2.2 and R2.4, the work still lacks the necessary in-depth analysis or material characterization to confirm the proposed mechanism. Without microscopic evidence it is difficult to distinguish whether the observed effect arises from intrinsic

broken symmetry, interfacial disorder, or oxide/diffusion effects. Such evidence is therefore essential for publication in a journal of this level.

Response 2.4: As we responded above, investigating the definite origin of this effect is not the central focus of this work. We have clearly demonstrated that this effect is present when different interface preparations, which are commonly used for superconducting qubit fabrication, are present. Regardless of the mechanism, it will play a role in affecting the performance of superconducting qubits, just like two-level systems.

R2.10: (1) In the PRX paper cited, dielectric loss was intentionally introduced, and therefore the resonator Q is not representative of intrinsic limits. If the authors claim that their resonators have Q-factors similar to those on doped substrates, then it is important to clarify what other mechanisms could limit their resonator Q. In addition, the APL paper cited analyzes the participation ratio of a qubit in a 3D cavity. Since qubits are subject to additional loss channels beyond those of resonators, a straightforward comparison of quality factors is not appropriate. A discussion of other possible loss mechanisms, such as trapped flux, seems necessary. (2) In Fig. S3, the S21 fits do not accurately capture the dips in the low-power curves (g–k). Since the S21 linewidth directly determines the resonator quality factor, these discrepancies are critical. Consequently, the results presented in Fig. S3(l) should be treated with particular caution and discussed more carefully.

Response 2.5: (1) The comparison we made is between samples fabricated on undoped silicon. While Ref. [31] primarily focused on the effect of doping, undoped samples were also included there as control cases (see Fig. 2b of Ref. [31]). We are not certain which APL paper the reviewer is referring to. It might be Ref. [6], *Phys. Rev. Appl.* 20, 014018 (2023)? In any case, we did not intend to draw direct comparisons with the results in this work. For the paper itself, Fig. 1b clearly shows that piezoelectric loss dominates the qubit lifetime, as evidenced by the qubit loss rate dropping by three orders of magnitude when coupled to a piezoelectric transducer.

R2.11: I am not familiar with the nonlinearity induced by TLS as proposed, and I ask the authors to clarify or confirm this hypothesis. It is well known that thin Al films doped with oxygen exhibit kinetic inductance, and a base pressure of 2×10^{-7} Torr is not sufficient to guarantee ultra-high-purity films. Moreover, a high resonator quality factor does not, by itself, confirm film purity, since it reflects only the distribution of dielectric loss contributions. If the authors wish to attribute the observed piezoelectricity to broken symmetry, they must provide direct evidence of film quality and purity.

Response 2.6: TLSs possess intrinsic nonlinearity, as their energy levels do not resemble those of harmonic oscillators. When the TLS excitation rate becomes comparable to its relaxation rate, the cavity loss induced by the TLS saturates. The aluminum films we studied indeed host small but finite kinetic inductances; however, the presence of kinetic inductance alone does not give rise to a piezoelectric response. Moreover, piezoelectricity remains present at elevated temperatures even after superconductivity is suppressed. Finally, we emphasize that, as mentioned above, we did not intend to attribute the observed piezoelectricity to *ideal* symmetry breaking, which, per the reviewer’s definition, happens only at ideal material interfaces. Inversion symmetry is broken at material interfaces regardless of the presence of native oxide or other disorder effects.

R2.15: The “parallel-plate” qubits I referred to are those reported in Nat. Commun. 12, 1779 (2021). In the authors’ response, this geometry seems to have been referred to as “planar rectangular.” This geometry represents some of the best reported T1 times in superconducting qubits (300 μ s, and more recently above 1 ms). To my knowledge, the planar circular geometry has not reached such a high T1. It is therefore surprising that in Fig. R1, the participation ratios for planar rectangular and planar circular geometries appear to be the same (though the plot is not very clear to me—please correct me if I am mistaken). I am particularly interested in how the simulations were performed. Since the piezoelectric layer thickness h is only up to 50 nanometers while the metallization features extend over hundreds of microns, the system presents an extreme aspect ratio of order 10^5 . In principle, a full 3D model is necessary to capture the geometry accurately, but such simulations face significant numerical challenges in resolving both scales simultaneously. By contrast, a 2D cross-sectional model may artificially yield identical participation ratios for planar rectangular and planar circular geometries, obscuring possible differences.

Response 2.7: The participation ratio provides useful guidance for estimating the potential T1 of a transmon qubit; however, it is not the only factor that determines the measured lifetime. Even for devices with identically designed geometries, T1 can vary significantly due to differences in fabrication processes, material quality, and surface treatments. The planar circulator geometry explored here is relatively unstudied compared to the conventional rectangular planar design, and the reported T1 values should not be interpreted as the upper limit achievable in this geometry under optimized conditions. Full 3D piezoelectric simulations are computationally prohibitive due to their complexity. Instead, we estimate the interface participation ratio using electrostatic simulations of the electric field distribution over the entire domain to obtain the spatial energy density. Because the thickness of the interfacial layer of interest is much smaller than the lateral dimensions of the electrodes, it is reasonable to assume that the electric field does not vary significantly along the out-of-plane (z) direction. This approximation allows an accurate

estimation of the energy stored in the piezoelectric layer and, consequently, the corresponding participation ratio.

For publication in a journal with high impact, the manuscript requires more comprehensive and in-depth analysis to identify and distinguish the microscopic mechanisms behind the observed piezoelectric behavior in Al/Si bilayers. At a minimum, the authors should determine whether the observed effects originate from broken symmetry, interfacial disorder, or film impurities. Without such evidence, I cannot recommend acceptance.

Response 2.8: As we have noted previously, while identifying the microscopic mechanism is indeed valuable, we respectfully disagree with the reviewer's assessment that significant progress can only be achieved through a full microscopic understanding. Our work establishes, for the first time, clear experimental evidence that piezoelectric coupling at Al/Si interfaces contributes to qubit energy loss, which is an effect that persists regardless of the specific microscopic origin of the interfacial polarization. We believe that elucidating and quantifying this coupling represents a meaningful and broadly relevant advance, even as the detailed microscopic mechanisms remain an important topic for future investigation.

Reviewer #3:

The responses to my comments in the rebuttal file seem mostly adequate. However, it seems like most of the changes that are mentioned there are not actually implemented in the updated manuscript file we received. For example, response #3.2-3.5, 3.6, 3.9, 3.11, and 3.20. Perhaps there was some mistake and the wrong version of the file was provided? Either way, in some cases I cannot judge if my concerns were completely addressed (for example response #3.20), since the rebuttal refers to additional text that doesn't exist.

Response 3.1: We inadvertently submitted a version of the manuscript that did not fully address all of the reviewers' comments. We sincerely apologize for this oversight and for any inconvenience it may have caused. We have now carefully reviewed the manuscript and attached the corrected version, with all changes clearly highlighted. For clarity, modifications made in the first revision are marked in blue, and those introduced in the current version are marked in orange. The relevant changes that were not included in the previous submission are listed below:

3.2: We have added Fig. S3 to the Supplementary Information to discuss the time-gating procedure. Additionally, we have added a paragraph to the Methods section.

3.3: We revised the relevant text. (Line 105 - 114)

3.4: We revised the relevant text. (Line 133 - 138, Line 368 - 369)

3.5: This comment was addressed in the previous version. The original data were shown in Fig. S8e-f. The data processing method is discussed in the caption of Fig. S8f.

3.6: We revised the relevant text. (Line 203 - 205)

3.9: We added the discussion to the manuscript. (Line 349-357)

3.11: We revised the relevant text. (Line 10)

3.15: We added the relevant text (Line 78-80) and Fig. S2.

3.19: We have added a sentence in the Supplementary Materials to justify the choice of dimensions used in the qubit simulations. (Line 745 - 746)

3.20: We revised the relevant text. (Line 437 - 447)

Reviewer #4:

Response 4.1: We thank the reviewer for co-reviewing the manuscript.